# A small RNA that cooperatively senses two stacked metabolites in one pocket for gene control

Griffin M. Schroeder [1,2], Chapin E. Cavender [1,2], Maya E. Blau [3], Jermaine L. Jenkins [1,2], David H. Mathews [1,2] & Joseph E. Wedekind [1,2]✉

Riboswitches are structured non-coding RNAs often located upstream of essential genes in bacterial messenger RNAs. Such RNAs regulate expression of downstream genes by recognizing a specific cellular effector. Although nearly 50 riboswitch classes are known, only a handful recognize multiple effectors. Here, we report the 2.60-Å resolution co-crystal structure of a class I type I preQ$_1$-sensing riboswitch that reveals two effectors stacked atop one another in a single binding pocket. These effectors bind with positive cooperativity in vitro and both molecules are necessary for gene regulation in bacterial cells. Stacked effector recognition appears to be a hallmark of the largest subgroup of preQ$_1$ riboswitches, including those from pathogens such as *Neisseria gonorrhoeae*. We postulate that binding to stacked effectors arose in the RNA World to closely position two substrates for RNA-mediated catalysis. These findings expand known effector recognition capabilities of riboswitches and have implications for antimicrobial development.

[1] Department of Biochemistry & Biophysics, University of Rochester School of Medicine & Dentistry, Rochester, NY 14642, USA. [2] Center for RNA Biology, University of Rochester School of Medicine & Dentistry, Rochester, NY 14642, USA. [3] University of Rochester, 120 Trustee Road, Rochester, NY 14627, USA. ✉email: joseph.wedekind@rochester.edu

Riboswitches are found primarily in the 5′ leader sequences of bacterial mRNAs where they regulate the expression of genes by recognizing a cognate effector[1–3]. These RNA-control elements usually comprise two domains: an aptamer that recognizes a metabolite with high specificity and an expression platform that contains gene-regulatory sequences[1]. Upon ligand binding, the expression platform undergoes conformational changes that alter the accessibility of key regulatory regions, such as the Shine-Dalgarno sequence (SDS), which must be unobstructed to initiate translation[3]. Direct observation of ligand-mediated transitions in riboswitches has enriched our understanding of RNA allostery and folding[4–7]. Riboswitches are also promising antimicrobial targets due to their presence in numerous human pathogens[2] and the finding that riboswitch dysregulation can compromise bacterial virulence[8].

PreQ$_1$-I (class I) riboswitches are the founding group of bacterial gene regulators that control the cellular concentration of queuosine (Q)[9] (Fig. 1a)—a hypermodified 7-deazapurine nucleobase required for translational fidelity in mammals and bacteria[10–12]. Although Q is not essential in bacteria, Q deficiency is associated with slow mid-log growth[13], compromised

stationary-phase viability[11], and loss of virulence[14]. Previous preQ$_1$-I riboswitch structures revealed an H-type pseudoknot fold, which recognizes a single preQ$_1$ ligand that completes coaxial stacking between flanking helices, thus stabilizing the expression platform[15–18]. The small size and well-defined fold of this class have spurred investigations of its folding and dynamics[4,15,19–21], effector specificity[9,22], the ligand-free to bound-state transition[4,18,19,22,23] and targeting with drug-like molecules[24,25]. Multiple bacterial species exhibit 1:1 riboswitch-to-preQ$_1$ stoichiometry[15–18,22,24,26], which is the prevailing ligand-binding mode of most riboswitches[2].

Importantly, the latter preQ$_1$-I riboswitch analyses have considered relatively few sequences. Recent work further classified preQ$_1$-I riboswitches into three subgroups called types I-III[27]. Inspection of the associated consensus models reveals that types I and II adopt similar secondary structures (Fig. 1b & Supplementary Fig. 1a, b). Although preQ$_1$-I$_{II}$ (type II) sequences prefer adenosine before the cytidine specificity base, preQ$_1$-I$_I$ sequences prefer uracil followed by CUA in the 3′-expression platform[9,27]. This observation and the results we describe in this study suggest that all previously studied sequences are preQ$_1$-I$_{II}$ riboswitches.

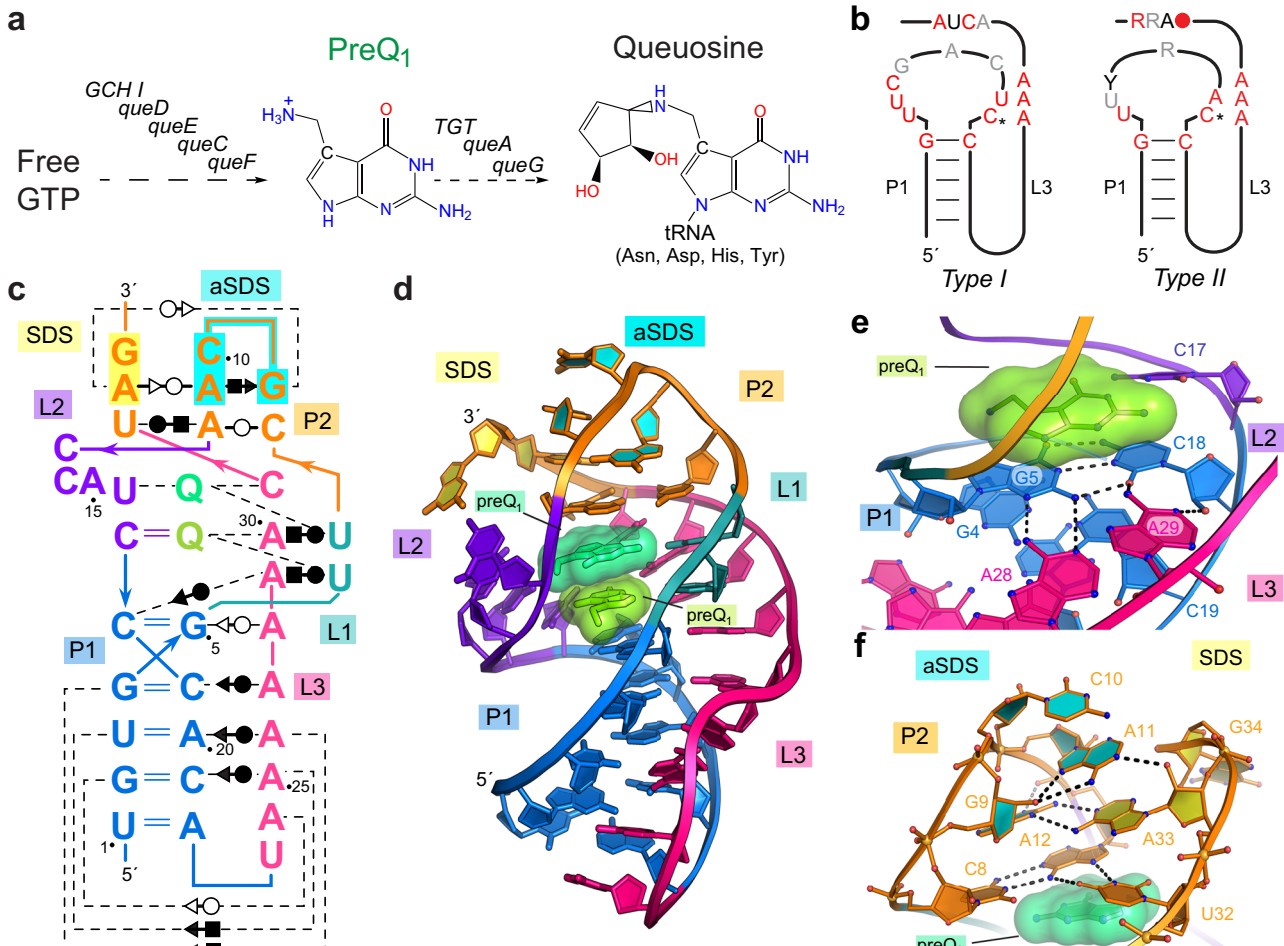

**Fig. 1 Queuosine biosynthesis, the preQ$_1$ riboswitch consensus model and co-crystal structure of the *Carnobacterium antarcticus* (*Can*) riboswitch.** **a** The queuosine (Q) biosynthetic pathway proceeds through the 7-deazapurine metabolite preQ$_1$[69]. **b** PreQ$_1$-I riboswitch subtypes shown as secondary structures based on covariation[27]. Red, black and gray positions indicate 97, 90, and 75% sequence conservation. Asterisk indicates a specificity base predicted to recognize preQ$_1$. **c** Secondary structure of the *Can* riboswitch. Colors correspond to specific pseudoknot base pairing (P) and loop (L) sequences. PreQ$_1$ is depicted as "Q". Noncanonical pairing is indicated by Leontis–Westhof symbols[70]. The Shine-Dalgarno sequence (SDS) and anti-(a) SDS are highlighted in yellow and cyan. **d** Ribbon diagram of the global *Can* riboswitch fold. **e** Binding pocket floor overview wherein floor bases comprise the A28•G5-C18 base triple. Dashed lines depict hydrogen bonds here and elsewhere. **f** Overview of the pocket ceiling, which comprises the U32•A12•C8 base triple. The view highlights P2 bases in the aSDS and SDS.

Importantly, preQ$_1$-I$_I$ riboswitches, found in gram-positive and -negative bacteria, are more represented than all other preQ$_1$ riboswitch subgroups and classes combined[27].

To elucidate the gene-regulatory properties of preQ$_1$-I$_I$ ribos-witches, we determined the co-crystal structure of a preQ$_1$-I$_I$ riboswitch from *Carnobacterium antarcticus*[28] (*Can*). The H-type pseudoknot structure unexpectedly reveals two bound preQ$_1$ effectors in a single aptamer (Fig. 1c, d). Although several riboswitches can recognize two effectors, these binding pockets are spatially separated[29–34]. In this respect, the *Can* preQ$_1$-I$_I$ riboswitch is exceptional because the metabolites stack tandemly, forming an unprecedented ligand-ligand interface within a single pocket. Using isothermal titration calorimetry (ITC) with in-house software that models two interdependent binding sites, we demonstrated that two preQ$_1$ effectors bind with positive coop-erativity. Mutants at each effector site reduce binding affinity and raise the concentration of preQ$_1$ required for gene repression in a bacterial reporter assay. We also found that additional preQ$_1$-I$_I$ sequences from *Haemophilus influenzae* (*Hin*) and *Neisseria gonorrhoeae* (*Ngo*) sense two preQ$_1$ effectors with positive cooperativity, suggesting that tandem, stacked effector binding is a hallmark of all preQ$_1$-I$_I$ riboswitches. Use of a single binding pocket to recognize two effectors has implications for the devel-opment of new antimicrobials that utilize a chemical scaffold that avoids cross-reactivity with naturally occurring metabolites.

## Results

**Features of the *Can* riboswitch fold**. To identify a suitable preQ$_1$-I$_I$ riboswitch for structural and functional analysis, we searched previously curated type I sequences[27] (Fig. 1b and Supplementary Fig. 1a) for a strong SDS (5′-AGGAG-3′) to use in a bacterial reporter assay[13]. We found several candidates, such as that from *Paenibacillus terrae*, but NCBI BLAST searches led to the discovery of an unreported sequence from *Can*[28]. This riboswitch crystallized readily from low salt solutions and the preQ$_1$-bound co-crystal structure was determined by molecular replacement. The structure was refined to 2.60 Å-resolution yielding $R_{work}/R_{free}$ values of 0.23/0.27 with acceptable quality-control metrics (Supplementary Table 1). Three crystal-lographically independent molecules were built, which showed varied quality in electron-density maps. Chains A and B are well defined, but the chain C P1-L3 junction shows a break (Supple-mentary Fig. 2a). Importantly, both effectors and the core apta-mer are well-resolved in each chain (Supplementary Fig. 2b), providing a firm foundation to guide functional experiments.

The overall fold of the *Can* preQ$_1$-I$_I$ riboswitch is an H-type pseudoknot (Fig. 1c, d). P1 is a canonical A-form helix whose minor groove is recognized by six A-amino-kissing interactions donated by the A-rich patch in L3 (Supplementary Fig. 3). This stabilizing segment culminates with a A28•G5-C18•A29 base-triple variation that forms the binding pocket floor and is reminiscent of preQ$_1$-I$_{II}$ riboswitch structures from *Thermo-anaerobacter tengcongensis* (*Tte*)[16,18,22] and *Bacillus subtilis* (*Bsu*)[15,17] (Fig. 1e). The pocket ceiling comprises an C8•A12•U32 base triple derived entirely from P2 (Fig. 1f). This configuration contrasts with preQ$_1$-I$_{II}$ riboswitches, in which the ceiling is formed by bases from both P2 and the L2 loop[9,16,17]. The preference for C8 and U32 in preQ$_1$-I$_I$ riboswitches appears to be incompatible with the base quadruple ceiling observed in preQ$_1$-I$_{II}$ riboswitches that require an adenine immediately before the cytosine specificity base (Fig. 1b & Supplementary Figs. 1a, b).

P2 also contains the expression platform, wherein the Watson-Crick (WC) face of A33 of the SDS pairs non-canonically with G9 and its 2′-hydroxyl interacts with the WC face of A11 (Fig. 1f). These SDS-anti(a)SDS interactions presumably attenuate

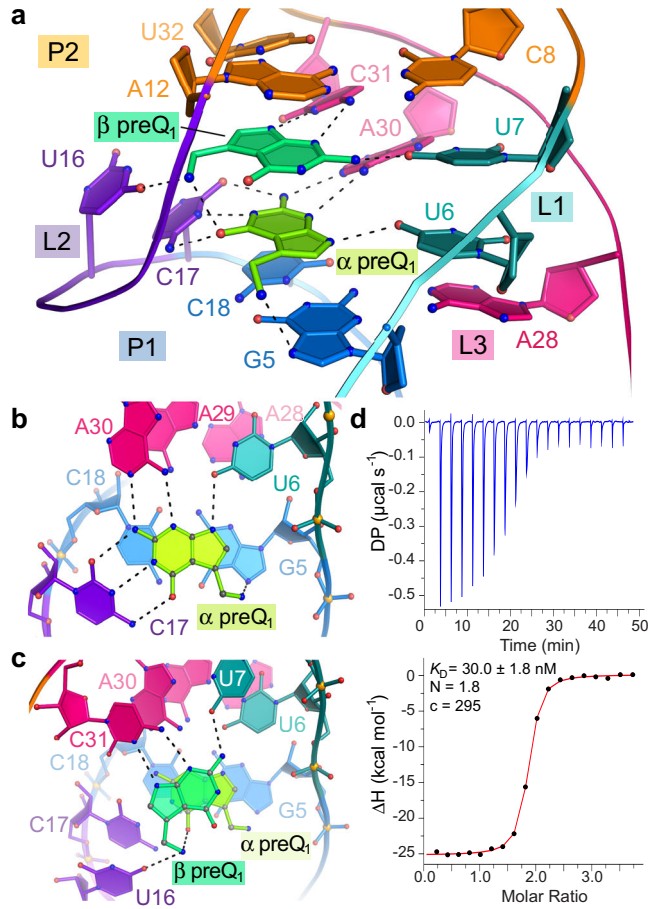

**Fig. 2 The *Can* preQ$_1$-I$_I$ riboswitch pocket with two preQ$_1$ ligands and confirmation of ligand-to-RNA stoichiometry. a** Overview of fully occupied binding pocket. Interactions in the (**b**) α site and (**c**) β site. **d** Representative ITC experiment with titration of preQ$_1$ into WT *Can* RNA. The binding constant $K_D$, ligand-to-RNA stoichiometry N, and c value are shown.

translation. Although we hypothesize that SDS nucleobase G34 makes a WC pair with C10, the former is involved in a crystal contact (Supplementary Fig. 2c). Notably, the *Tte* preQ$_1$-I$_{II}$ riboswitch forms the expected aSDS-SDS C-G intramolecular WC pair, while also exhibiting non-canonical pairing in its expression platform[16,18,22], as observed here for the *Can* preQ$_1$-I$_I$ riboswitch.

**Stacked metabolites in a small aptamer**. A distinguishing feature of our structure is two preQ$_1$ molecules, which we term α and β, stacked in a single aptamer pocket (Fig. 2a). Recognition at the α site is conserved among preQ$_1$-I$_I$ and preQ$_1$-I$_{II}$ riboswitches, wherein specificity is conferred by a cytidine that recognizes preQ$_1$ by a *cis* WC interaction. Other conserved α-site con-tributions include the WC face of A30, the major-groove edge of U6 and the major-groove edge of G5, which interacts with the preQ$_1$ methylamine (Fig. 2b & Supplementary Figs. 4a–c). In contrast, the β site has not been observed previously. Bases C31 and U7, which are highly conserved among type I sequences, confer specificity for preQ$_1$ by contributing three hydrogen bonds that recognize the metabolite edge (Fig. 2c & Supplementary Fig. 1a). The β-site preQ$_1$ interacts with the α-site effector through aromatic stacking and donation of a hydrogen bond from the methylamine to both the keto oxygen of the α-site effector and O4 of U16 (Fig. 2a, c). The mode of β-site effector recognition differs from all known preQ$_1$ riboswitches, including

preQ$_1$-II[35] and preQ$_1$-III[36], which utilize *trans* WC-pairing to read the preQ$_1$ face (Supplementary Figs. 4d, e). Although other riboswitches bind two effectors, these examples involve distinct binding pockets that spatially separate the ligands[32–34,37]. To our knowledge, recognition of two interacting ligands in a single aptamer pocket is unprecedented in RNA biology.

**Stacked recognition is cooperative**. Interacting ligands should cooperatively influence each other's binding. Analysis of the *Can* riboswitch by ITC at 25 °C showed that the wildtype (WT) sequence binds preQ$_1$ with an average macroscopic $K_D$ of 32.0 ± 2.0 nM and a ligand-to-receptor ratio (N) of 1.8 (Fig. 2d). Fitting to a single-phase isotherm supports binding with positive cooperativity, in accord with our structure. Enthalpy drives binding and offsets the predicted entropic cost of ordering two ligands, producing a favorable $\Delta G°$ (Supplementary Table 2). Analysis at 37 °C to accentuate cooperative binding produced a parabolic thermogram best described by a binding model wherein two interdependent ligands occupy non-equivalent sites (Supplementary Figs. 5a, b). We implemented this model to assess the macroscopic binding constant of each interaction, which yielded $K_{D1}$ of 891 nM and $K_{D2}$ of 461 nM for the first and second binding events. The improved affinity observed for the second preQ$_1$ relative to the first indicates positive cooperativity, exemplified by the macroscopic cooperativity constant, γ, of 7.7 (Supplementary Table 3).

We next generated *Can* riboswitch mutants to probe recognition at the α and β sites. Position 17 is a major determinant of α-site specificity (Fig. 2a, b) and the C17U mutation severely weakened binding as indicated by macroscopic $K_{D1}$ and $K_{D2}$ values of 3.13 μM and 1.30 μM (Supplementary Fig. 5c & Supplementary Table 3). This result is consistent with the position of C17 in our structure and an equivalent nucleobase in the *Bsu* preQ$_1$-I$_I$ aptamer[9]. C17U showed a parabolic isotherm suggesting retention of two binding events. Likewise, position 31 shows a prominent role in β-site specificity (Fig. 2a, c). The C31U mutation produced a comparable parabolic isotherm, corresponding to $K_{D1}$ and $K_{D2}$ values of 6.64 μM and 10.26 μM (Supplementary Fig. 5d & Supplementary Table 3). As expected from the structure, C31U severely affects β-site recognition. Both C17U and C31U retain positive cooperativity with γ values of 9.6 and 2.6 (Supplementary Table 3).

**Dual-binding signatures in other preQ$_1$-I$_I$ riboswitches**. C17 and C31 are highly conserved in the type I consensus model (Fig. 1b & Supplementary Fig. 1a) and the importance of each is confirmed by our structure and ITC experiments. We next asked if dual binding is evident in other type I riboswitches. We used ITC at 25 °C to evaluate sequences from *Hin* and *Ngo*, which belong to the *Proteobacteria* phyla rather than the *Firmicutes* (Supplementary Fig. 1a). The *Hin* riboswitch binds preQ$_1$ with a $K_D$ of 52.9 ± 0.2 nM whereas the *Ngo* riboswitch binds with a $K_D$ of 50.5 ± 1.3 nM; like WT *Can*, each binds with an N of ~2 (Supplementary Table 2 & Supplementary Figs. 5e, f). Analysis at 37 °C accentuates the cooperative character of isotherms (Supplementary Figs. 5g, h), resulting in γ values of 26.7 and 32.9 that indicate substantial positive cooperativity for each (Supplementary Table 3). We note a high degree of sequence identity exists in the binding pocket of the *Can*, *Hin* and *Ngo* riboswitches (Supplementary Fig. 1a). Significantly, each possesses key nucleobases required for α and β site preQ$_1$ recognition including U6, U7, U16, C17, A30, and C31 (Supplementary Fig. 6). Given ITC evidence of cooperativity for all three riboswitches, it appears that each riboswitch uses a similar mode of dual, stacked preQ$_1$ recognition.

**Gene regulation requires two effectors**. Using a GFP*uv* reporter gene[13,38] controlled by the *Can* riboswitch in live cells, we asked whether both preQ$_1$ molecules were required for effective gene regulation (Fig. 3a). We hypothesized that when both sites are occupied the SDS would be less accessible, leading to greater repression of GFP*uv* translation (Fig. 3b); likewise, intermediate levels of translation would occur if one site is occupied. Dose-response analysis of the WT riboswitch produced a biphasic curve with EC$_{50}$ values of 96 ± 14 nM (EC$_{50,\ 1}$) and 7100 ± 360 nM (EC$_{50,\ 2}$) (Fig. 3c & Supplementary Table 4). Collectively, both binding events confer 15.4-fold repression, comparable to the 14.9-fold repression observed for the *Lactobacillus rhamnosus* (*Lrh*) preQ$_1$-II riboswitch[35], which binds a single ligand with an EC$_{50}$ of 15 nM[13] (Fig. 3c, **inset**, 3d & Supplementary Table 4). Notably, the *Can* riboswitch sensing range is broader than the *Lrh* riboswitch in this assay, suggesting that it detects preQ$_1$ over a wider range of effector concentrations. At present, the basis for this apparent sensing difference is uncertain (see below). To ensure that the changes in GFP*uv* expression were riboswitch driven, we evaluated a positive control containing an SDS without an upstream riboswitch and a negative control lacking the SDS[13]. As expected, neither control responded to changes in preQ$_1$ concentration (Fig. 3c, d). In accord with ITC data, C17U and C31U mutants each showed poorer EC$_{50}$ values that were ~60-fold higher and ~210-fold higher than WT (Fig. 3c–e & Supplementary Table 4). While each mutant retains dual binding in vitro, the elevated EC$_{50}$ values imply that preQ$_1$ levels must be significantly higher inside cells to elicit an efficient gene-regulatory response, underscoring the importance of each effector binding site for gene regulation.

Although our data cannot differentiate a preferred order of preQ$_1$ binding, impairment of the β site had a more pronounced effect on gene regulation (Fig. 3d, e). While C17U elicited a sixfold repression, the C31U variant repressed GFP*uv* expression by only twofold (Fig. 3d, e). This functional disparity—also reflected by poorer C31U $K_{D1}$ and $K_{D2}$ values (Supplementary Table 3)—could be due to the requirement of the β effector to serve as a scaffold that supports the binding pocket ceiling via stacking (Fig. 1f). In this manner, the β site orders P2 in the gene off state while binding at the α site either orders the β site pocket or stabilizes effector binding at the β site.

## Discussion

We described the structure and cooperative binding of a small riboswitch that senses two stacked effectors in a single binding pocket. Examination of all known preQ$_1$-I sequences encompassing multiple phyla revealed that nucleobases that compose the α and β binding sites are conserved only within preQ$_1$-I$_I$ sequences (Supplementary Fig. 1a). In contrast, only nucleobases associated with α site recognition are conserved within preQ$_1$-I$_{II}$ sequences, consistent with known *Tte* and *Bsu* riboswitches structures (Supplementary Figs. 1b, 4b, c, & 6) and previous bioinformatic analysis[27]. Although experimental analysis of the preQ$_1$-I$_{III}$ riboswitch is sparse, it appears that nucleobases associated with α site recognition are conserved in preQ$_1$-I$_{II}$ representatives, but not those associated with β site recognition (Supplementary Fig. 1c). This is consistent with previous ITC experiments, which demonstrated that this riboswitch binds with a 1:1 stoichiometry[39]—like preQ$_1$-I$_{II}$ representatives. Accordingly, the unprecedented mode of dual effector recognition appears to be a hallmark of the most common and taxonomically diverse preQ$_1$ riboswitch group[2,27], the preQ$_1$-I$_I$ riboswitch, which has been overlooked until now.

Cooperative riboswitches are posited to show a steep "digital" dose-response[40], yet the *Can* riboswitch exhibits a broad, biphasic

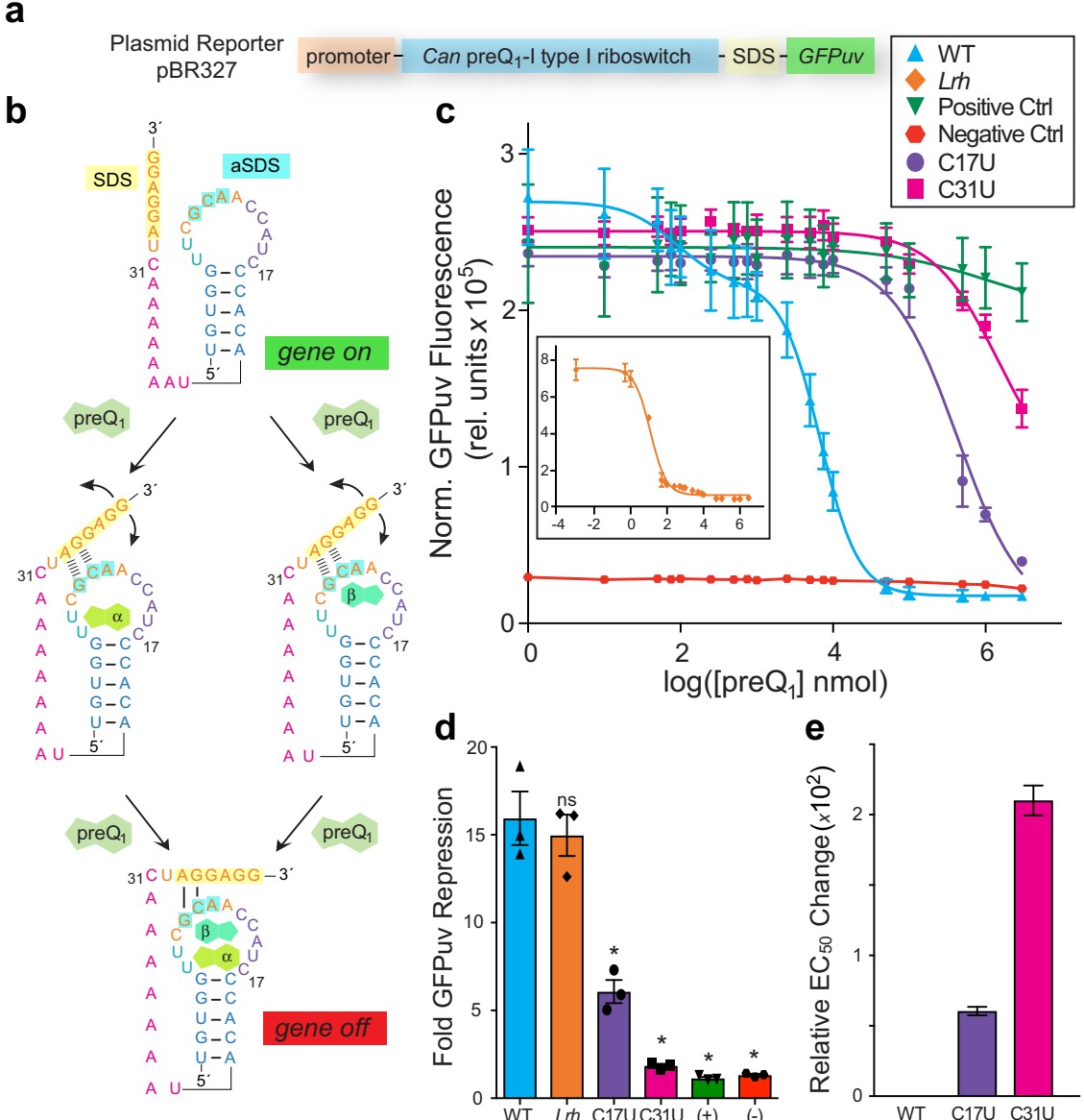

**Fig. 3 Riboswitch reporter assay and dose response in live bacteria. a** Schematic of the plasmid reporter. **b** Two-site binding model wherein preQ$_1$ can bind either site first. **c** Average GFP$uv$ emission dependence on preQ$_1$; (inset) one-site binding by the $Lrh$ preQ$_1$ riboswitch[38]. **d** Bar graph showing fold repression of GFP$uv$ emission for the $Can$, $Lrh$ and mutant riboswitches with individual points shown. **e** Bar graph showing fold change in average EC$_{50}$ relative to $Can$ riboswitch EC$_{50,2}$. Significance was determined by a two-tailed Student's $t$ test with Welch's Correction ($n = 3$ biological replicates. *$p \leq 0.05$). S.E.M. is shown in **c** and **d**; propagated errors are shown in **e**.

dose-response in our in-cell GFP$uv$ assay despite the positive cooperativity we observe in our ITC analysis (Supplementary Table 3). Although it is tempting to associate each transition in our GFP$uv$ assay with an individual preQ$_1$ binding event, the intracellular concentration of preQ$_1$ is not known in such assays and depends on multiple factors, such as the efficacy of 7-deazapurine transporters[41,42]. Additionally, we cannot rule out possible competition between preQ$_1$ and other metabolites in the cellular milieu[43], as observed for the $glmS$ riboswitch[44]. These, or other factors, likely influence the shape of the $Can$ riboswitch dose-response curve (Fig. 3c); nevertheless, the preQ$_1$-I$_I$ riboswitch is expected to maintain positive cooperativity inside the cell[43].

Our data allow us to conclude that dual-effector recognition is critical for efficient gene regulation by preQ$_1$-I$_I$ riboswitches—as indicated by the deleterious effects caused specific α and β site

mutants. However, we can only speculate on the reason why cooperativity evolved in preQ$_1$-I$_I$ riboswitches but not in other types or classes of the preQ$_1$ riboswitch family. Our data suggest that the level of regulation attained is similar between the preQ$_1$-I$_I$ $Can$ riboswitch and the preQ$_1$-II $Lrh$ riboswitch, despite differences in preQ$_1$ binding stoichiometry[35] (Fig. 3d). This result suggests that these two disparate riboswitch folds evolved equally effective chemical networks to sense a common effector for gene regulation. Yet, cooperativity is expected to provide notable benefits in regulation efficiency. One such advantage is that gene expression is permitted when metabolite levels are low (Fig. 3b, *middle panel*), while assuring the ability to quickly attenuate expression before excess effector accrues in the cell[43]. This is reasonable considering that many preQ$_1$-I$_I$ riboswitches control the translation of transporters that salvage Q-precursor metabolites from the extracellular environment[9,27,45].

Although the *Can* and *Lrh* riboswitches differ in terms of binding stoichiometry and overall fold, each positions its expression platform near the binding pocket. This organization raises the question of whether dual, stacked metabolite binding could be effective to regulate folds wherein the aptamer is located distally from the expression platform. PreQ₁-III riboswitches exemplify this organization, wherein the expression platform can be as far as 40 Å away from the aptamer[36]. Communication between the single-effector binding pocket and an orthogonal SDS-antiSDS helix is mediated by an A-minor base that makes a T-shaped contact with the edge of preQ₁[36] (Supplementary Fig. 4e). The preQ₁-II riboswitch uses a similar pocket[35] and the A-minor base was shown to be essential for gene-regulatory function[38]. It is conceivable that dual, stacked effector recognition could be used by the preQ₁-III riboswitch fold, if the effectors promoted coaxial helical stacking, and one or both ligands were detected by the A-minor motif. Accordingly, we predict that additional riboswitches that bind dual, stacked effectors exist in nature.

Extant riboswitches can also provide clues about the organization of extinct ribozymes[46]. Riboswitches that utilize distal binding domains to accommodate a single ligand suggest how the folds of early ribozymes were organized to position substrates[46]. Our findings extend this concept to single-domain ribozymes. In particular, the *Can* aptamer shows how a ribozyme could position two substrates in one pocket to promote covalent bond formation. Intriguingly, the α-site primary amine is solvent accessible (Supplementary Fig. 7), providing a key functional group absent from the RNA chemical repertoire[47]; in contrast, the β-preQ₁ WC face is solvent accessible. Notably, O6-methyl preQ₁ shows site-specific preQ₁-I_II riboswitch methylation[48], providing a precedent for ligand-mediated chemical transformation of RNA. These observations collectively suggest how a ribozyme could position two substrates within a single compact fold to facilitate chemistries required for prebiotic metabolism.

PreQ₁-I_I riboswitches are prominent in human pathogens[9,27] including *Ngo*, an urgent public-health threat[49]. The mode of effector recognition by the preQ₁-I_I riboswitch provides new opportunities to target such regulatory RNAs. For example, a single small molecule that simultaneously occupies both α and β binding sites could reduce cross-reactivity with targets that recognize preQ₁-like molecules (e.g., guanine), yielding greater potency and reduced toxicity. Our results suggest that such riboswitches merit further exploration for their potential as antimicrobial targets.

## Methods

**Data reporting**. No statistical methods were used to predetermine sample size. The experiments were not randomized and the investigators were not blinded to allocation during experiments and outcome assessment.

**RNA purification**. RNA strands were synthesized by Dharmacon (Lafayette, CO) as described by the manufacturer except that deprotection heating was 30 min at 65 °C. RNA was purified by 15% denaturing PAGE and DEAE chromatography[50]. DEAE buffer was replaced with 0.02 M Na-HEPES pH 6.8, 0.10 M ammonium acetate, and 0.002 M EDTA; care was taken to minimize UV exposure[51]. After ethanol precipitation of pooled DEAE fractions, RNA was dissolved in Nanopure™ UV/UF (ThermoFisher) water and desalted on a PD-10 column (GE Healthcare). Quality was assessed by analytical PAGE stained with SYBR Gold (Thermo-Fisher) and visualized on a GelDoc (BioRad XR +). The yield was measured spectrophotometrically. Lyophilized RNA was stored at −20 °C.

**Structure determination**. Lyophilized RNA was dissolved in 20 μL of 0.01 M sodium cacodylate pH 7.0 and concentrated to 800 μM by centrifugation. Separate volumes of the concentrated riboswitch and an equal volume of folding buffer (0.004 M MgCl₂, 0.01 M sodium cacodylate pH 7.0 and 0.0016 M preQ₁) were heated at 65 °C for 3 min. The folding mix was added dropwise to the RNA and heated 3 min at 65 °C, followed by slow cooling to 24 °C.

Crystals were grown from VDX plates (Hampton Research) by hanging-drop vapor-diffusion. A 1 μL volume of RNA was combined with 1 μL of precipitant drawn from 1 mL in the well. Crystals grew from solutions of 30% (v/v) 2-methyl-2,4-pentanediol, 0.08 M KCl, 0.012 M NaCl, 0.04 M sodium cacodylate pH 5.5, and 0.002 M hexammine cobalt (III) chloride. Crystals grew in 3 weeks at 20 °C as hexagonal rods of size 0.125 mm × 0.040 mm × 0.040 mm. Crystals were cryo-protected by 2 min transfers into well solution supplemented with 40% to 60% (v/v) 2-methyl-2,4-pentanediol. Single rods were captured in nylon loops using 16 mm copper pins (Hampton Research) with the *c*\*-axis oriented parallel to the φ axis. Crystals were plunged into N₂(l) for shipping to the Stanford Synchrotron Radiation Lightsource (SSRL).

X-ray data were collected remotely on beamline 12-2 using Blu-Ice software and the Stanford Auto-Mounter[52,53] at a λ of 0.9800 Å with a Δφ of 0.15°, an exposure time of 0.7 s per image with 450 total images, and a sample-to-detector distance of 425 mm at 100 K. All data were recorded on a PILATUS 6 M detector (Dectris Inc). Data-collection strategies were generated using Web-Ice[54]. Diffraction data were reduced with autoxds[55] using XDS, as well as CCP4 programs POINTLESS, AIMLESS and TRUNCATE[56,57]. The structure was determined by molecular replacement in PHENIX[58] starting from the *B. subtilis* preQ₁-I riboswitch (Protein Data Bank entry 3FU2). The top solution for three molecules in the asymmetric unit produced a TFZ of 9.2 and a log-likelihood gain of 289. The structure was built in COOT with additional refinement in PHENIX[58]. Intensity and refinement statistics are in Supplementary Table 1. Cartoons, schematic diagrams and surface renderings of coordinates were generated in PyMOL (Schrödinger LLC). In Supplementary Fig. 7, preQ₁ atoms were colored by solvent accessible surface area using the color area (solvent) function in PyMOL (Schrödinger LLC). The reported solvent accessible surface area were calculated in PISA[59] (PDBe PISA v1.52) for chain A, as implemented in CCP4[57].

**Isothermal titration calorimetry**. Each sample was folded by dissolving lyophilized RNA in 250 μL 0.01 M sodium cacodylate pH 7.0. RNA was heated to 65 °C for 3 min and mixed with an equal volume of preheated folding buffer at 65 °C comprising 0.01 M sodium cacodylate pH 7.0 and 0.004 M MgCl₂. The combined solution was heated for an additional 3 min, then slow cooled to 24 °C followed by overnight dialysis against 2 L of ITC buffer (0.050 M Na-HEPES pH 7.0, 0.10 M NaCl and 0.004 M MgCl₂) using a 3500 MWCO Slide-A-Lyzer Dialysis Cassette G2 (Thermo-Scientific). PreQ₁ from a 0.020 M stock in water was diluted to 0.0010 M in ITC buffer.

ITC was conducted using two different instruments. Experiments with WT *Can*, *Ngo* and *Hin* riboswitches at 25 °C were conducted on a PEAQ-ITC (Malvern) with RNA in the cell and preQ₁ in the syringe over 19 injections. Experiments were carried out with an injection volume of 4 μL (0.5 μL technical injection) and a spacing of 150 s. These thermograms were analyzed with MicroCal PEAQ-ITC Analysis software (Malvern Panalytical, Inc) using a 'single-sites' binding model, which corresponds to the independent sites model below.

To obtain additional data points for cooperativity analysis, WT experiments were also conducted at 37 °C on a VP-ITC (MicroCal). Experiments were carried out with an injection volume of 10 μL (6 μL technical injection) and a spacing of 240 s with RNA in the cell and preQ₁ in the syringe over 29 injections. Mutant riboswitches were analyzed similarly but at 25 °C due to poor binding. These thermograms were analyzed using a 'two-interdependent non-equivalent sites' model (Supplementary Fig. 5b) as described below.

In each case, at least two measurements were performed for each RNA sample on the appropriate instrument. Representative thermograms and curve fits are provided in Supplementary Fig. 5. Thermodynamic parameters for experiments performed on the PEAQ ITC are in Supplementary Table 2 and experiments on the VP-ITC are in Supplementary Table 3. Macroscopic ΔG° values for mutant riboswitches represent the sum of microscopic ΔG° values, which were obtained by calculating $K_{rel}$ at each binding event versus the WT riboswitch at 25 °C. The concentrations of RNA and preQ₁ used in ITC experiments are reported in the source data file.

**Least-squares regression analysis of ITC experiments (two interdependent non-equivalent sites model)**. ITC experiments performed on the VP instrument produced parabolic thermograms indicative of cooperativity but these could not be satisfactorily fit with conventional ITC software as noted[37]. Structural evidence indicates that the preQ₁ ligands interact in their respective binding pockets, suggesting that a cooperative binding model in which the two effector-binding sites are nonequivalent and interdependent was appropriate. We implemented this model (Supplementary Fig. 5b) in a custom Python program based on the binding polynomial theory[60].

Rather than fitting an apparent stoichiometry, we fixed the number of binding sites to exactly two and fit a nuisance parameter that represents the effective concentration of active riboswitch RNA in the ITC cell relative to the recorded concentration[60,61]. Although the binding model describes a binding enthalpy and a microscopic dissociation constant for each of four distinct binding equilibria (Supplementary Fig. 5b), there are only three independent microscopic dissociation constants:

$$K_{D,A1} = \frac{[R][L]}{[RL_A]} \quad K_{D,B1} = \frac{[R][L]}{[RL_B]} \quad K_{D,A2} = \frac{[RL_B][L]}{[RL_{AB}]} \quad K_{D,B2} = \frac{[RL_A][L]}{[RL_{AB}]} \quad (1)$$

$$K_{D,A1}K_{D,B2} = K_{D,B1}K_{D,A2} = \frac{[R][L]^2}{[RL_{AB}]} \quad (2)$$

Likewise, there are only three independent binding enthalpies because enthalpy is a state function; completing a thermodynamic cycle must result in no enthalpy change.

$$\Delta H^{\circ}_{A1} + \Delta H^{\circ}_{B2} - \Delta H^{\circ}_{A2} - \Delta H^{\circ}_{B1} = 0 \quad (3)$$

The binding polynomial results in a cubic equation in the concentration of free ligand $[L]$

$$[L]^3 + (2R_T - L_T + K_{D,A2} + K_{D,B2})[L]^2 + ((R_T - L_T)(K_{D,A2} + K_{D,B2})$$
$$+ K_{D,A1}K_{D,B2})[L] - L_T K_{D,A1}K_{D,B2} = 0 \quad (4)$$

where $R_T = [R] + [RL_A] + [RL_B] + [RL_{AB}]$ is the total concentration of RNA in the ITC cell and $L_T = [L] + [RL_A] + [RL_B] + 2[RL_{AB}]$ is the total concentration of preQ$_1$ in the ITC cell. We solved this cubic equation analytically by choosing the root that satisfies $[L] = 0$ when $L_T = 0$.

Inspired by several approaches[62–64], we explicitly accounted for the dilution of all chemical species present due to displacement of the liquid in the ITC cell by the injection volume. The differential changes in the concentrations of bound species due to a differential injected volume $dV$ are

$$d[RL_A] = \frac{1}{V_0}(-[RL_A]dV + d\Phi_{A1} - d\Phi_{B2})$$
$$d[RL_B] = \frac{1}{V_0}(-[RL_B]dV + d\Phi_{B1} - d\Phi_{A2}) \quad (5)$$
$$d[RL_{AB}] = \frac{1}{V_0}(-[RL_{AB}]dV + d\Phi_{A2} + d\Phi_{B2})$$

where $V_0$ is the volume of the ITC cell and $\Phi_i$ is the flux through the binding equilibrium $i$. The enthalpy can be expressed as a function of the total injected volume $V$

$$H(V) = \frac{1}{V_0}\left(\Delta H^{\circ}_{A1}\frac{d\Phi_{A1}}{dV} + \Delta H^{\circ}_{B1}\frac{d\Phi_{B1}}{dV} + \Delta H^{\circ}_{A2}\frac{d\Phi_{A2}}{dV} + \Delta H^{\circ}_{B2}\frac{d\Phi_{B2}}{dV}\right) \quad (6)$$

The enthalpy change associated with a particular injection that brings the stoichiometric ratio of ligand to receptor $S$ from $S_{i-1}$ to $S_i$ is given by the average value of the enthalpy over this interval.

$$\Delta H_i = \frac{1}{S_i - S_{i-1}}\int_{S_{i-1}}^{S_i} dSH(S) \quad (7)$$

Inserting Eqs. 1–3 and Eqs. 5–6 into Eq. 7 and using integration by parts gives the injection enthalpy change in terms of the fit parameters, the ITC cell volume, the initial concentrations of RNA in the ITC cell $R_0$ and of ligand in the syringe $L_0$, and the concentration of free ligand obtained as the solution to Eq. 4.

$$\Delta H_i = \frac{\Omega(S_i) - \Omega(S_{i-1})}{L_0(S_i - S_{i-1})}$$

$$\Omega(S) = \frac{\left(\frac{L_0}{R_0} + S\right)\left(L_T - [L]\right)\left(\Delta H_{A1}K_{D,B2} + \Delta H_{B1}K_{D,A2} + (\Delta H_{A1} + \Delta H_{B2})[L]\right)}{K_{D,A2} + K_{D,B2} + 2[L]} \quad (8)$$

We used a trust-region reflective algorithm[65] implemented in the optimize.least_squares() method of SciPy[66] to minimize the following cost function:

$$F(\vec{\theta}, \lambda) = \sum_{i=1}^{N}\left(\Delta H_i(\vec{\theta}) - \Delta H_{i,obs}\right)^2 + \lambda\sum_{j=1}^{M}\left(\frac{\theta_j - \theta_{j,0}}{w_j}\right)^2 \quad (9)$$

where $N$ is the number of observed injections and $M$ is the number of fit parameters. The first term is a least-squares term describing the goodness-of-fit between the estimated and observed injection enthalpy changes. The second term is an L2 regularization term—whose relative strength is controlled by the hyperparameter $\lambda$—that prevents overfitting by penalizing deviations of the fit parameters $\theta$ from a target value $\theta_0$. In a Bayesian framework, this penalty is interpreted as a Gaussian prior on the fit parameters with mean $\theta_0$ and standard deviation $w$[67]. For the three independent microscopic dissociation constants, regularization was applied to the natural logarithm of the dissociation constant. The regularization targets were set to the values of the fit parameters from a binding model assuming two independent and equivalent binding sites, i.e. the model used by most commercial ITC software. We derived analytical derivatives of the cost function given by Eq. 9 with respect to the fit parameters to take advantage of computationally efficient gradient-based optimization methods.

For each RNA sequence, we performed a global fit to obtain a single set of fit parameters informed by multiple experiments in which the initial concentrations of riboswitch receptor and ligand vary in order to interrogate different regions of the resulting thermogram. One offset parameter, a constant added to the estimated injection enthalpy changes, was fit for each experiment. The hyperparameter $\lambda$ controlling the relative strength of the regularization term was optimized for each RNA sequence individually by cross validation across experiments. Each

experiment was fit individually for a sequence of $\lambda$ with logarithmic spacing —$\log_{10}\lambda$ was varied from $-6$ to $+6$ in steps of 0.125. The resulting fit parameters were used to estimate the value of the cost function for the other experiments involving the same RNA sequence. The value of $\lambda$ with the smallest average value of the cost function for experiments not used to train the parameters was chosen for the global fit.

Two sets of values were used for the regularization weights $w$. For $Can$ WT, $Can$ C31U and $Hin$ WT, the regularization weights were $16k_BT$ for the binding enthalpies and log10 for the logarithms of the dissociation constants, where $k_B$ is the Boltzmann constant and $T$ is the absolute temperature. However, these weights produced poor quality fits for $Can$ C17U and $Ngo$ WT, as revealed by fit parameters with large bootstrapped uncertainties. As such, the regularization weights for these sequences were $16k_BT$ for both the binding enthalpies and the logarithms of the dissociation constants. For all RNA sequences, the regularization weights were 1 kcal mol$^{-1}$ for the offsets and 0.05 for the nuisance parameter describing the effective RNA concentration.

To derive estimates and 95% confidence intervals for the fit parameters, we used a bootstrapping method to resample the fitting target in the nonlinear regression[68]. In each bootstrap iteration, we added the residual from the initial fit multiplied by a random number sampled from a standard normal distribution to the observed injection enthalpy changes. The resulting distributions of fit parameters are non-normal, and so we report the estimate of each fit parameter as the median of the bootstrap parameter distribution. We also report a 95% confidence interval as the (2.5, 97.5) percentiles of this bootstrap distribution.

In addition to the fit parameters, we also calculate the following derived parameters: cooperativity $C$, macroscopic dissociation constants $K_{D,1}$ and $K_{D2}$, and a macroscopic cooperativity $\gamma$.

$$C = \frac{K_{D,A1}}{K_{D,A2}}$$

$$K_{D,1} = \left(\frac{1}{K_{D,A1}} + \frac{1}{K_{D,B1}}\right)^{-1}$$

$$K_{D,2} = K_{D,A2} + K_{D,B2}$$

$$\gamma = \frac{4K_{D,1}}{K_{D,2}} \quad (10)$$

We obtained estimates and 95% confidence intervals for derived parameters by calculating the derived parameters for each bootstrap iteration and then reporting the median and (2.5, 97.5) percentile of the bootstrap parameter distribution.

**In-cell GFPuv reporter assay.** The WT $Can$ riboswitch was placed into the pBR327-$Lrh$(WT)-GFPuv plasmid upstream of the GFPuv reporter gene (Fig. 3a). Riboswitch mutants were prepared by site-directed mutagenesis (GenScript Inc.) on the WT sequence, which were verified by DNA sequencing. For experiments involving the $Lrh$ riboswitch, the parent pBR327-$Lrh$(WT)-GFPuv plasmid was used[13].

The assay was performed as described[13,38] with some exceptions. E. coli strain JW2765 $\Delta queF$ cells (Coli Genetic Stock Center, Yale University)—incapable of preQ$_1$ biosynthesis—were transformed with the desired plasmid and grown on CSB agar plates containing both ampicillin (100 μg mL$^{-1}$) and kanamycin (50 μg mL$^{-1}$). Single colonies were isolated to inoculate overnight liquid cultures of 3 mL CSB-amp-kan media. These were used to inoculate 2 mL of fresh CSB-amp-kan media with varying concentrations of preQ$_1$: 0, 1 nM, 10 nM, 50 nM, 75 nM, 100 nM, 250 nM, 500 nM, 750 nM, 1 μM, 2.5 μM, 5 μM, 7.5 μM 10 μM, 50 μM, 100 μM, 500 μM, 1 mM, and 3 mM; the highest concentration corresponds to the solubility limit of preQ$_1$ in CSB[13]. Three or more biological replicates were measured for each concentration. All measurements and analysis were performed as described[13] using Prism (GraphPad Software, Inc). The replicates in each construct were compared using the "compare datasets" function before analysis. The WT $Can$ curve showed a biphasic model whereas others were best described by a log(inhibitor) dose versus response (three parameters).

An unpaired student's t-test with a Welch's correction was used to analyze fold repression data (Fig. 3d). The $p$ value for WT $Can$ vs. WT $Lrh$ was 0.6429 ($t = 0.503$, degrees of freedom (df) = 3.76, 95% confidence interval = -4.515–6.453). The $p$ value for WT $Can$ vs. C17U $Can$ was 0.0125 ($t = 5.94$, df = 2.72, 95% confidence interval = -15.47 to -4.266). The $p$ value for WT $Can$ vs C31U $Can$ was 0.0112 ($t = 9.23$, df = 2.02, 95% confidence interval = -20.61 to -7.583). The $p$ value for WT $Can$ vs. the negative control was 0.0106 ($t = 9.59$, df = 2.01, 95% confidence interval = -21.16 to -8.076). The $p$ value for WT $Can$ vs the positive control was 0.0103 ($t = 9.71$, df = 2.01, 95% confidence interval = -21.3400 to -8.2800).

Notably, fluorescence emission in the absence of preQ$_1$ is comparable between all riboswitch constructs and the positive control; moreover, the WT $Can$ and $Lrh$ sequences repress GFPuv fluorescence emission to a level comparable to the negative control —demonstrating the rigor of the assay (Supplementary Fig. 8). An unpaired student's t-test with a Welch's correction was also used to analyze fluorescence emission data. The $p$ value for WT $Can$ was 0.0037 ($t = 14.96$,

df = 2.09, 95% confidence interval = -173452 to -98511). The $p$ value for C17U *Can* was 0.0175 ($t$ = 6.64, df = 2.18, 95% confidence interval = -212164 to -53149). The $p$ value for C31U *Can* was 0.1263 (t = 2.05, df = 3.22, 95% confidence interval = -107817 to -21313). The $p$ value for WT *Lrh* was 0.0006 (t = 33.92, degrees of freedom (df) = 2.15, 95% confidence interval = -437151 to -344271). The $p$ value for the positive control was 0.7441 (t = 0.35, df = 3.45, 95% confidence interval = -61691 to 48527). The $p$ value for the negative control was 0.0631 (t = 2.60, df = 3.80, 95% confidence interval = -11377 to 488.8).

**Reporting Summary**. Further information on research design is available in the Nature Research Reporting Summary linked to this article.

## Data availability

The data supporting the findings of this study are available from the corresponding authors upon reasonable request. Structure factor amplitudes and coordinates for the *Can* preQ$_1$-I$_I$ riboswitch were deposited in the Protein Data Bank under accession code 7REX. Publicly available PDB entries used in this study are: 6VUI, 3FU2, 4RZD, and 4JF2. Source data includes injection data for ITC, and fluorescence emission and cell growth readings for in-cell assays. Source data are provided with this paper.

## Code availability

The ITC fitting software and parameter fits are available on GitHub at https://github.com/chapincavender/itc_two_site_fit, distributed under the MIT license.

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

## Acknowledgements

We thank members of the Wedekind and Mathews labs for technical assistance and helpful discussions. We thank Dr. P. Dumas for binding model inspiration based on his unpublished work cited herein. Support for this research was provided by the National Institutes of Health National Institute of General Medical Sciences (NIH NIGMS) grants R01 GM132185 to D.H.M and R01 GM063162 to J.E.W. G.M.S. was supported by training grant T32 GM118283, and C.E.C was supported by training grant T32 AI049815 to J.E.W. and an Elon Huntington Hooker graduate fellowship. Use of the Stanford Synchrotron Radiation Lightsource, SLAC National Accelerator Laboratory, is supported by the U.S. Department of Energy, Office of Science, Office of Basic Energy Sciences under Contract No. DE-AC02-76SF00515. The SSRL Structural Molecular Biology Program is supported by the DOE Office of Biological and Environmental Research, and by NIH NIGMS (P30 GM133894). The contents of this publication are solely the responsibility of the authors and do not necessarily represent the official views of NIH or NIGMS.

## Author contributions

GMS, CEC, DHM, and JEW designed experiments. GMS, MEB performed experiments. GMS, CEC, JLJ, DHM, and JEW analyzed the data. GMS, CEC, and JEW wrote the paper. All authors reviewed the manuscript.

## Competing interests

The authors declare no competing interests.
