## [Peer Review File · Nature Communications]

Title: A small RNA that cooperatively senses two stacked metabolites in one pocket for gene controlREVIEWER COMMENTS

Reviewer #1 (Remarks to the Author):

Wedekind and colleagues present a highly novel crystal structure of a type-I preQ-1 riboswitch that reveals the binding of two ligand that directly interact with one another. In support of their structural analysis, the authors present a detailed calorimetric analysis of cooperative binding in several representative type-I preQ-1 riboswitches to make the case that this unusual mode of ligand binding is highly likely to be a central characteristic of this grouping. Finally, a cell based reporter assay demonstrates that ablation of either ligand binding site causes a significant shift in the extracellular preQ-1 concentration required to elicit the regulatory response. Together, the authors present a solid structure-function analysis that reveals a new mode of ligand recognition in RNA.

Overall, this manuscript is well-written and provides clear and mostly compelling data (see below, with respect to the cell-based assay). The crystallographic and calorimetric data is rigorously analyzed and the authors present a convincing set of conclusions. The figures are also clear and highlight key points in the text well. This work is highly worthy of publication, once the authors address a set of minor points, as follows.

Lines 76-77. I somewhat disagree with the statement “Although several riboswitches can recognize two effectors, these sites are positioned in separate domains”. This essentially comes down to how the authors define “domain” in RNA. I tend to think of a domain as an independently folding element, such as the classic P4-P6 domain of the T. tetrahymena group I intron. In that light some of the two ligand binding riboswitches are single folding domains, such as the THF riboswitch. This RNA clearly folds as a single domain with the main folding center around the 3WJ site that enables the formation of the pseudoknot site. I really don’t think that this detracts from the author’s main point, and they should consider modifying this claim (also made on line 137).

Line 183. Given the nature of the cell-based reporter assay, I would argue that this is a two significant figure experiment rather than three (i.e., 90 ± 3 nM). Also, I am confused by the assertion that the two phases correspond to a preQ1 binding event based upon the ITC data. What do the authors specifically envision each state to be? I assume they think that the second transition is a single ligand event? If so, why would this event elicit the greatest repression (as opposed to the first event, which contributes moderately). I admit that I am a bit confused by the observed biphasic transition given the degree of cooperativity in binding.

Line 188 – 189. The statement that the riboswitch sensing “acts a ‘dimmer’ switch rather than a ‘digital’ switch” is predicated on the assumption that the concentration of ligand added to the medium is the same as in the cell, which is not known by the authors. For some metabolites, high affinity influx pumps can significantly concentrate the compound and the shape of the curve can reflect the behavior of that component of metabolism. The authors should use caution in interpreting their cell-based data. In the discussion the authors also reiterate this conclusion (lines 206 – 207). I would like to see a more

concrete discussion as to why the authors think that a positively cooperative system leads to a less steep response curve, since this observation is counter to what was hypothesized.

Figure 3. In interpreting the first phase of the transition, the error bars seem quite large and overlap between the two baselines of that transition. I am surprised given the size of the errors in this transition that the stated error is as low as cited.

Reviewer #2 (Remarks to the Author):

The manuscript of Schroeder et al describes the structure of a type I preQ1 riboswitch from *Carnobacterium antarcticus* as well as extensive biophysical characterization to corroborate the structural findings. The main finding is that the aptamer domain of this riboswitch binds two stacked preQ1 molecules, something that had not been observed before. Strong binding data support a cooperative binding model and in vivo data corroborate the importance of the two binding model. Overall, this is a very strong manuscript. The data are of excellent quality and the structure provides new information that helps understand preQ1 riboswitches better. The observation of two binding ligands is novel for riboswitches and helps support several important ideas regarding evolution of function in the RNA world. There are a few minor points that need to be addressed:

1. The authors conclude that binding of two preQ1 molecules is a hallmark of all type I preQ1 riboswitches. While the evidence presented is good, I think a couple of mutagenesis experiments to support it would enhance the manuscript. Similar experiments to the ones done with the Can riboswitch to show the involvement of C31 and U17 but using a different organism would help address this point.
2. A multiple sequence alignment of many preQ1 riboswitches should be added to the Supplemental Materials to highlight the common features.
3. Additional discussion on the differences amongst the three types is needed, highlighting what was learned from the current work.
4. Extended Data Figure 4 is very confusing. Panels mix with each other. The figure needs to be redesigned for additional clarity.

Reviewer #3 (Remarks to the Author):

In this manuscript, Schroeder and co-workers report a detailed structural, biochemical, and biological study describing a Class I preQ1 riboswitch from *Carnobacterium antarcticus* (and providing information more broadly about this class of riboswitch). Remarkably, their crystallographic studies reveal that the riboswitch binds two copies of the preQ1 metabolite, exhibiting positive cooperativity, and stacking directly next to each other in the binding site. Biological studies using a reporter system reveal a “dimmer switch”-type response to ligand over a wide concentration range. This work is notable as it is the first example of an RNA binding to 2 copies of a single metabolite, with implications for riboswitch

evolution, RNA catalysis and potentially synthetic biology as well. The manuscript is highly rigorous and well written and methodology is sound. References are appropriately cited. This work will be of broad interest and will definitely be appropriate for the readers of Nature Communications. I have several minor comments:

1. The title is somewhat misleading. There are other examples of riboswitches that bind 2 different metabolites, but this is notably the first example of a switch that binds 2 molecules of the same metabolite (in the same binding pocket!). More precise language would better emphasize the important advance reported here – I suggest “A small RNA senses its effector in a tandem stacked mode for cooperative gene control” or something similar.

2. It is somewhat non-obvious that the system exhibits positive cooperativity but the response is increased in concentration range (a “dimmer switch”). This is counter to the typical example where positive cooperativity generally results in an increased Hill coefficient (and a reduced range of effect, as noted in the discussion). Can the authors comment further? Note, the Hill coefficient model was developed for enzymes, which are of course a fundamentally different system on several levels. Since this is the first example of a riboswitch that senses 2 copies of a single molecule it has fundamental importance and would be worth discussing for readers outside this specific field interested in this unique biochemical phenomenon.

3. In the case of this example, the SDS is partially embedded within the aptamer domain. However, in Class III preQ1 switches, the SDS is found further outside. Do the authors think such dual binding events as observed here are uniquely effective when the SDS is within the aptamer? This might be worth adding a sentence or two in the discussion.

To be clear, these very minor comments do not dampen enthusiasm for what I consider to be a very strong and rigorous study. I strongly support the publication of this manuscript once the issues above have been addressed.

We thank each of the reviewers for providing detailed and insightful comments to improve the quality of our manuscript. We are especially gratified by the uniformly positive evaluations of the work. Reviewer #1 stated, "... *this manuscript is well-written and provides clear and mostly compelling data... highly worthy of publication ...*". Reviewer #2 wrote, "... *this is a very strong manuscript. The data are of excellent quality and the structure provides new information*". Reviewer #3 wrote, "[the] *manuscript is highly rigorous and well written and methodology is sound ... This work will be of broad interest...*". Moreover, each of the reviewers stated that the work would be suitable for publication following "*minor*" revisions. We have positively addressed all comments as requested with one exception (noted below). As a result, we believe that the manuscript has improved significantly and that the work is now ready for publication.

To track our changes, we created the ensuing point-by-point response. Our responses are in red with specific changes highlighted in yellow (reviewer #1), green (reviewer #2) or cyan (reviewer #3). A marked-up manuscript with color coding for each reviewer is appended to this document. We have also provided our complete written responses following each reviewer's questions (below).

Our noteworthy revisions are summarized as follows:

- i.* The title was changed to be clearer, as requested by rev. #2
- ii.* The abstract was shortened and references were removed to be consistent with the journal style. The Extended Data were moved to Supplementary Information to be consistent with the journal style.
- iii.* A more circumspect approach was taken when interpreting the GFP μ v assay based on the critiques of rev. #1 and rev #3. Specifically, the interpretation of the in-cell assays as evidence for a dimmer switch response was removed in light of the limitations of this assay (noted in the revision) and positive cooperativity observed in our ITC analysis.
- iv.* We produced a new multisequence alignment as Supplementary Figure 1 based on comments from rev. #2. We described the sequence hallmarks that differentiate the various preQ $_1$ -I riboswitch types in light of the consensus model, the new multisequence alignment and known co-crystal structures.
- v.* We added a new discussion about the prospect that other riboswitches use dual stacked effector recognition, as suggested by rev. #3.
- vi.* Based on our own reading of the manuscript, we found and corrected errors in Fig 1c regarding the highlighted aSDS. We also added a missing H-bond in Fig. 1e (and removed underlying H-bonds for clarity). We corrected an error in the base pairing of Fig. 3b, which now shows the correct SDS-aSDS interaction. We added individual data points to the bar plots in Fig. 3d to match the journal style. We clarified the average K_D value reported for WT *Can* in the text and the representative metrics shown in 2d. These parameters were updated in Supplementary table 2. Other self-identified minor typos are colored red but remain unhighlighted.
- vii.* We included all of our raw data from ITC and GFP μ v assays in a single spreadsheet, as per journal policy.
- viii.* We made minor corrections to the Supporting Table 1 of refinement statistics that we discovered during the final stages of PDB deposition.

- ix. We made changes to the Methods to match journal policy and to describe the treatment of errors for EC₅₀ fitting, based on comments from rev. #1.

The detailed point-by-point responses to reviewers are as follows:

Reviewer #1 (Remarks to the Author):

Lines 76-77. I somewhat disagree with the statement “Although several riboswitches can recognize two effectors, these sites are positioned in separate domains”. This essentially comes down to how the authors define “domain” in RNA. I tend to think of a domain as an independently folding element, such as the classic P4-P6 domain of the *T. tetrahymena* group I intron. In that light some of the two ligand binding riboswitches are single folding domains, such as the THF riboswitch. This RNA clearly folds as a single domain with the main folding center around the 3WJ site that enables the formation of the pseudoknot site. I really don’t think that this detracts from the author’s main point, and they should consider modifying this claim (also made on line 137).

We agree that the THF riboswitch uses one folding unit. In terms of effector binding, we also agree that two spatially separated sites (i.e., a 3-way junction and a pseudoknot region) are used to recognize individual effectors. However, our rationale was that each site fits the definition of a ‘domain’ as described by [Pley *et al.* (1994) *Nature* **372**, 68-74]. In this manuscript, McKay and co-workers state, “a hammerhead RNA-DNA ribozyme-inhibitor complex at 2.6 Å resolution reveals that the base-paired stems are A-form helices and that the core has two structural domains. The first domain is formed by the sequence 5'-CUGA following stem I and is a sharp turn identical to the uridine turn of transfer RNA, whereas the second is a non-Watson-Crick three-base-pair duplex with a divalent-ion binding site”. There is a detailed description of *domain I* and *domain II* in this paper. As such, this represents a somewhat different perspective about a domain. Obviously, unlike the THF riboswitch, the *Can* riboswitch’s binding sites are not spatially separated. We simply wanted to convey the latter point, but we see how our nomenclature could be confusing to the audience. The reviewer’s concept of a domain is also more rigorous and is widely accepted by the structural biology community.

To avoid ambiguities, we changed our text to state (new lines 75-78) “Although several riboswitches can recognize two effectors, these **binding pockets are spatially separated**³⁰⁻³⁵. In this respect, the *Can* preQ₁-I₁ riboswitch is exceptional because the metabolites stack tandemly, forming an unprecedented ligand-ligand interface **within a single pocket**.” These changes avoid the definition of a “domain” while clarifying our meaning.

Similarly, we changed line 137: “Although other riboswitches bind two effectors, these examples involve distinct **binding pockets** that spatially separate the ligands^{4, 6, 7, 43}”

On Line 139 we added: “recognition of two interacting ligands in a single aptamer **pocket** is unprecedented in RNA biology.”

Line 216 of the Discussion. We clarified, “We described the structure and cooperative binding of a small riboswitch that senses two stacked effectors **in a single binding pocket**.”

Line 183. Given the nature of the cell-based reporter assay, I would argue that this is a two significant figure experiment rather than three (i.e., 90 ± 3 nM).

We agree that two significant figures are more appropriate. We have changed all EC_{50} measurements reported in the text (e.g., line 190 & 193) to reflect this point. In **Supplementary Table 4**, we changed the number of significant figures to two as well.

Also, I am confused by the assertion that the two phases correspond to a preQ1 binding event based upon the ITC data. What do the authors specifically envision each state to be? I assume they think that the second transition is a single ligand event? If so, why would this event elicit the greatest repression (as opposed to the first event, which contributes moderately). I admit that I am a bit confused by the observed biphasic transition given the degree of cooperativity in binding.

Given the concern of reviewer #1 and #3 on this topic and comments below, we have taken a more circumspect interpretation of the two-phase dose response curve. Accordingly, we removed suggestions that the two-phase binding curve represents two discrete binding events of the preQ₁ metabolite, which appears to be an overinterpretation of the data. We thank the reviewer for pointing this out. Accordingly, we have made the following changes to the text:

Lines 81-83. We deleted, “~~Unexpectedly, the Can preQ₁-I₁ riboswitch showed an extended effector sensing range that is more akin to a dimmer switch than a digital switch^{27,36}.~~”

Lines 80-81. We added the following text instead: “Mutants at each effector site reduce binding affinity and raise the concentration of preQ₁ required for gene repression in a bacterial reporter assay.”

Line 192-193. To clarify the binding by the preQ₁-II riboswitch, “for the *Lactobacillus rhamnosus* (Lrh) preQ₁-II riboswitch³⁷, which binds a single ligand with an EC_{50} of 15 nM¹³”.

Lines 195-197. We changed the text to be more cautious, “Notably, the Can riboswitch sensing range is broader than the Lrh riboswitch in this assay, suggesting that it detects preQ₁ over a wider range of effector concentrations. At present, the basis for this apparent sensing difference is uncertain (see below).”

Line 200-204. We removed emphasis on the growth curve, “In accord with ITC data, C17U and C31U mutants each showed poorer EC_{50} values that were ~60-fold higher and ~210-fold higher than WT (Figs. 3c-e & Supplementary Table 4). While each mutant retains dual binding in vitro, the elevated EC_{50} values imply that preQ₁ levels must be significantly higher inside cells to elicit an efficient gene-regulatory response, underscoring the importance of each effector binding site for gene regulation.”

The observation that one phase of the biphasic WT curve seems to impart more gene regulatory activity than the other is a keen one. Although we agree that we cannot directly relate our *in vitro* ITC data to our cell assay, our structure suggests that the β effector could provide a platform on which the ceiling can stack, thus ordering the P2 region and stabilizing the gene-off state. This prediction is supported by the observation that the C31U mutant regulates gene expression more poorly than C17U.

Lines 205-212: We added a passage that relates our in-cell mutant data to our structure. We wrote, “Although our data cannot differentiate a preferred order of preQ₁ binding, impairment of the β site had a more pronounced effect on gene regulation (Figs. 3d,e). While C17U elicited a 6-fold repression, the C31U variant repressed GFP_{uv} expression by only 2-fold (Figs. 3d,e). This functional disparity — also reflected by poorer C31U K_{D1} and K_{D2} values (Supplementary

Table 3) — could be due to the requirement of the β effector to serve as a scaffold that supports the binding pocket ceiling via stacking (**Fig. 1f**). In this manner, the β site orders P2 in the gene off state while binding at the α site either orders the β site pocket or stabilizes effector binding at the β site.”

We addressed the interpretation of the biphasic curve in more detail in the next point.

Line 188 – 189. The statement that the riboswitch sensing “acts a ‘dimmer’ switch rather than a ‘digital’ switch” is predicated on the assumption that the concentration of ligand added to the medium is the same as in the cell, which is not known by the authors. For some metabolites, high affinity influx pumps can significantly concentrate the compound and the shape of the curve can reflect the behavior of that component of metabolism. The authors should use caution in interpreting their cell-based data. In the discussion the authors also reiterate this conclusion (lines 206 – 207). I would like to see a more concrete discussion as to why the authors think that a positively cooperative system leads to a less steep response curve, since this observation is counter to what was hypothesized.

We see the reviewer’s point and believe it is worth mentioning in the main text. As such, we revised the text to more circumspectly consider the GFPuv assay results and the positive cooperativity measured by ITC. We also added a statement that competition between preQ₁ and other metabolites could affect the shape of the dose-response curve. We noted that other riboswitches interact with other metabolites in a cellular context, as reported for the *glmS* riboswitch/ribozyme [Watson, P. Y. & Fedor, M. J. *Nat. Struct. Mol. Biol.* **18**, 359-363 (2011)].

Lines 229-238 state:

“Cooperative riboswitches are posited to show a steep “digital” dose-response²
^{42, 43}, yet the *Can* riboswitch exhibits a broad, biphasic dose-response in our in-cell GFPuv assay despite the positive cooperativity we observe in our ITC analysis (**Supplementary Table 3**). Although it is tempting to associate each transition in our GFPuv assay with an individual preQ₁ binding event, the intracellular concentration of preQ₁ is not known in such assays and depends on multiple factors, such as the efficacy of 7-deazapurine transporters^{44,45}. Additionally, we cannot rule out possible competition between preQ₁ and other metabolites in the cellular milieu⁴⁶, as observed for the *glmS* riboswitch⁴⁷. These, or other factors, likely influence the shape of the *Can* riboswitch dose-response curve (**Fig. 3c**); nevertheless, the preQ₁-I₁ riboswitch is expected to maintain positive cooperativity inside the cell⁴⁶.”

Figure 3. In interpreting the first phase of the transition, the error bars seem quite large and overlap between the two baselines of that transition. I am surprised given the size of the errors in this transition that the stated error is as low as cited.

This is an excellent observation and we see the reviewer’s point. **Fig. 3c** depicts the average of three datasets with the standard error of the mean shown for each point. In response to the reviewer’s question, we contacted GraphPad for insight into how best to address this question and whether we used an appropriate approach. The representative recommended that we employ the “compare datasets” function to assess whether each replicate in a dataset should be analyzed in separate columns or together in sub-columns using GraphPad. Indeed, the different modes of analysis alter how the software calculates the errors. In the case of the WT *Can* datasets, the software recommended analyzing each dataset separately. For the other constructs, the software recommended analyzing all three replicates together in sub-columns.

As a result, we now report the standard error between the three separately determined EC₅₀ values for WT *Can*. (Line 190) The new EC₅₀ parameters for the WT *Can* construct using two significant figures are:

EC_{50,1} = 96 ± 14 nM (previously 86 ± 3 nM)

EC_{50,2} = 7100 ± 360 nM (previously 6800 ± 200 nM)

Although the values did not change appreciably, the errors did increase and are more in line with what is expected from the curves shown in **Fig. 3c**. The graph shown in **Fig. 3c** remains the same even though these new errors are used.

We also changed the EC_{50,1} and EC_{50,2} values in **Supplementary Table 4** and recalculated the Fold EC₅₀ change metrics reported in Fig. 3e based on this comment

Lines 473-475: We added brief description of the GraphPad consideration in the Methods, “*The replicates in each construct were compared using the “compare datasets” function before analysis.*”

Reviewer #2 (Remarks to the Author):

1. The authors conclude that binding of two preQ₁ molecules is a hallmark of all type I preQ₁ riboswitches. While the evidence presented is good, I think a couple of mutagenesis experiments to support it would enhance the manuscript. Similar experiments to the ones done with the *Can* riboswitch to show the involvement of C31 and U17 but using a different organism would help address this point.

We appreciate the reviewer’s suggestion for more experiments; however, we believe that sufficient evidence already exists based upon our current analysis and the addition of a new multisequence alignment (recommended by reviewer #2 in the next point). Our rationale is that the cost-to-benefit ratio for such new experiments will be large given the time and resources involved. Specifically, these experiments are not standard ITC experiments because they require large amounts of RNA and preQ₁ for the VP-ITC due to the poor *K_D* values of mutants, as well as the need to capture the full parabolic character of cooperative binding for analysis by our Python program.

As the reviewer noted, “*the evidence presented is good*” and we believe such experiments would be merely incremental. The reviewer’s suggestion of a multisequence alignment combined with the existing consensus model strongly bolsters our conclusion that all type I preQ₁ riboswitches appear to use dual, stacked effector recognition. Indeed, nucleotides that engage in preQ₁ binding at the *Can* riboswitch α and β sites are 97% conserved based on all known type I sequences (>1,500 representatives) [McCown, P. J., Liang, J. J., Weinberg, Z. & Breaker, R. R. *Chem Biol* **21**, 880-889 (2014).].

As the reviewer observed, our existing experimental data firmly support the requirement for nucleotides C17 and C31 at the respective α and β sites for dual preQ₁ binding and function. With the benefit of our new *Can* preQ₁-I₁ riboswitch co-crystal structure, it is clear that all known type I riboswitch sequences possess these two bases — and other nucleotides — required for α and β site recognition. Please see new **Supplementary Fig. 1a**.

Moreover, our manuscript also demonstrates that the WT *Can*, *Ngo* and *Hin* preQ₁-I₁ riboswitches —spanning multiple phyla — each bind two preQ₁ molecules based on ITC. All three species show the characteristic parabolic response to preQ₁ when the experiment is performed at 37 °C (**Supplementary Figs. 5a,g,h**), which is best described by a two-interdependent-sites binding model (**Supplementary Fig. 5b**) wherein the macroscopic cooperativity constants, γ , support positive cooperativity (**Supplementary Table 2**). Collectively the data suggest that the mode of binding is the same in all three sequences (**Supplementary Fig 6a**), and that this analysis extends to the entire type I subclass (**Supplementary Fig. 1a**).

2. A multiple sequence alignment of many preQ₁ riboswitches should be added to the Supplemental Materials to highlight the common features.

We agree with the reviewer and we have added a multiple sequence alignment in new **Supplementary Fig. 1**. We carefully selected phylogenetically diverse sequences from each type of class I preQ₁ riboswitch to provide the greatest diversity — albeit type III is found almost exclusively in gamma proteobacteria. These sequence alignments and the consensus models — which were derived from all known representatives analyzed by McCown *et al.* [McCown, P. J., Liang, J. J., Weinberg, Z. & Breaker, R. R. *Chem Biol* **21**, 880-889 (2014).] — collectively illustrate that the nucleobases involved in type I recognition are absolutely conserved but are absent in type II and type III preQ₁-I riboswitches.

To accentuate differences in the signature residues engaged in preQ₁ binding by the type I and type II preQ₁-I riboswitches, our new figure includes sequences from riboswitches that were crystallized previously (bolded genus and species). In the columns above each base, we denoted positions that contact each preQ₁ effector (bolded) and positions that form the P1 helix (underlined). The structural mapping upon the riboswitch sequences nicely explains the type I and type II covariation models [McCown, P. J., Liang, J. J., Weinberg, Z. & Breaker, R. R. *Chem Biol* **21**, 880-889 (2014).]. Importantly, the type I covariation model considers >1,500 sequences and our crystallographic data account for why specific bases are conserved (i.e., because they engage in α and β -site preQ₁ recognition). For the sake of brevity, we chose to sample a small number of diverse sequences, although the covariation models — also included in the figure — were derived from all sequence representatives from each riboswitch type [adapted from McCown *et al.* & Breaker (2014) *Chem & Biol* **21**, 880].

For type III riboswitches, there is no crystallographic data to indicate which nucleobases contact preQ₁ beyond the conserved cytidine in loop L2 — which presumably contacts the WC face of preQ₁ [McCown, P. J., Liang, J. J., Weinberg, Z. & Breaker, R. R. *Chem Biol* **21**, 880-889 (2014).]. However, the alignment and consensus model clearly indicate that nucleobases required for beta site preQ₁ recognition are absent. Moreover, we previously published ITC data from a representative type III riboswitch and found 1:1 binding stoichiometry [Lieberman, J. A., Bogue, J. T., Jenkins, J. L., Salim, M. & Wedekind, J. E. ITC analysis of ligand binding to preQ₁ riboswitches. *Meth. Enzymol.* **549**, 435-50 (2014).].

The new supplementary figure legend states:

Supplementary Figure 1 | Covariation model and multisequence alignments of preQ₁ class I riboswitches. (a) Type I Covariation models generated from the full group of known sequence representatives (adapted from ref. 2); red, black and gray positions indicate 97%, 90% and 75% sequence conservation. Multisequence alignments were generated using a handful of representatives derived from phylogenetically diverse bacteria (reported by McCown *et al.*²). Positions in bold within the alignment recognize preQ₁ based on the *C. antarcticus* co-crystal structure of this investigation. PreQ₁ binding nucleobases at the α and β sites are each denoted

in the covariation model and the sequence alignment as α or β . Here and elsewhere, bolded organisms have been structurally characterized (this work). In addition to the greatest number of representative sequences (indicated in italics), preQ₁-I_{II} riboswitches exhibit the greatest taxonomic diversity². (b) same as (a), but with type II sequences. Characterized sequences are *T. tencongensis*^{3, 4} and *B. subtilis*². Asterisks denote conserved preQ₁ recognition positions. (c) same as (a) and (b) but with type III sequences. Due to a lack of structural characterization, the canonical specificity base is the only predicted preQ₁ recognition position². Alignments were created in JALVIEW⁶.”

3. Additional discussion on the differences amongst the three types is needed, highlighting what was learned from the current work.

We agree with the reviewer, and have added the following paragraph to the first paragraph of our discussion:

The new text describing this analysis is on Lines 216-228:

“We described the structure and cooperative binding of a small riboswitch that senses two stacked effectors *in a single binding pocket*. Examination of all known preQ₁-I sequences encompassing multiple phyla revealed that nucleobases that compose the α and β binding sites are conserved only within preQ₁-I_{II} sequences (Supplementary Fig. 1). In contrast, only nucleobases associated with α site recognition are conserved within preQ₁-I_{II} sequences, consistent with known *Tte* and *Bsu* riboswitches structures (Supplementary Figs. 1b, 4b,c & 6) and previous bioinformatic analysis²⁰. Although experimental analysis of the preQ₁-I_{III} riboswitch is sparse, it appears that nucleobases associated with α site recognition are conserved in preQ₁-I_{II} representatives, but not those associated with β site recognition (Supplementary Fig. 1c). This is consistent with previous ITC experiments, which demonstrated that this riboswitch binds with a 1:1 stoichiometry⁴¹ — like preQ₁-I_{II} representatives. Accordingly, the unprecedented mode of dual effector recognition appears to be a hallmark of the most common and taxonomically diverse preQ₁ riboswitch group^{30, 35}, the preQ₁-I_{II} riboswitch, which has been overlooked until now.”

4. Extended Data Figure 4 is very confusing. Panels mix with each other. The figure needs to be redesigned for additional clarity.

We understand the reviewer’s concern. As requested, we reformatted the figure to clearly differentiate among the panels. In the revision, we rearranged the ITC panels, increased the spacing between them and made the schematic diagram larger, thus increasing the readability. The revised figure is presented as **Supplementary Fig. 5**.

Reviewer #3 (Remarks to the Author):

1. The title is somewhat misleading. There are other examples of riboswitches that bind 2 different metabolites, but this is notably the first example of a switch that binds 2 molecules of the same metabolite (in the same binding pocket!). More precise language would better emphasize the important advance reported here – I suggest “A small RNA senses its effector in a tandem stacked mode for cooperative gene control” or something similar.

We agree with the reviewer that the title (which must be fifteen-words or less) should convey better the novel mode of effector recognition. We believe we have captured this point with the new title:

A small RNA that cooperatively senses two stacked metabolites in one pocket for gene control

2. It is somewhat non-obvious that the system exhibits positive cooperativity but the response is increased in concentration range (a “dimmer switch”). This is counter to the typical example where positive cooperativity generally results in an increased Hill coefficient (and a reduced range of effect, as noted in the discussion). Can the authors comment further? Note, the Hill coefficient model was developed for enzymes, which are of course a fundamentally different system on several levels. Since this is the first example of a riboswitch that senses 2 copies of a single molecule it has fundamental importance and would be worth discussing for readers outside this specific field interested in this unique biochemical phenomenon.

In response to this comment and those from reviewer 1, we have altered our interpretation of the in-cell experiments. Upon reflection, we agree with rev. #1 that it is not possible to relate the observed positive cooperativity from ITC to the curve shapes derived from the GFP_{uv} reporter assay for the reasons s/he stated. There are several reasons for this:

1. We do not know the intracellular preQ₁ concentration.
2. We do not know the efficiency of associated 7-deazapurine influx transporters.
3. The experiment takes place in a complex cellular environment. In this scenario, preQ₁ likely competes with other metabolites for binding, altering the apparent concentration of preQ₁ needed to elicit a specific regulatory response.

Our response to reviewer #1 in this regard appears on Lines 231-241. We wrote:

^{42, 43} “Cooperative riboswitches are posited to show a steep “digital” dose-response², yet the Can riboswitch exhibits a broad, biphasic dose-response in our in-cell GFP_{uv} assay despite the positive cooperativity we observe in our ITC analysis (Supplementary Table 3). Although it is tempting to associate each transition in our GFP_{uv} assay with an individual preQ₁ binding event, the intracellular concentration of preQ₁ is not known in such assays and depends on multiple factors, such as the efficacy of 7-deazapurine transporters^{44,45}. Additionally, we cannot rule out possible competition between preQ₁ and other metabolites in the cellular milieu⁴⁶, as observed for the glmS riboswitch⁴⁷. These, or other factors, likely influence the shape of the Can riboswitch dose-response curve (Fig. 3c); nevertheless, the preQ₁-I₁ riboswitch is expected to maintain positive cooperativity inside the cell⁴⁶.”

However, as rev. #3 suggested, we also considered the benefits of positive cooperativity for gene regulation. This discussion provides background and context for the broader community.

On lines 239-251 of the Discussion we wrote,

“Our data allow us to conclude that dual-effector recognition is critical for efficient gene regulation by preQ₁-I₁ riboswitches — as indicated by the deleterious effects caused by specific α and β site mutants. However, we can only speculate on the reason why cooperativity evolved in preQ₁-I₁ riboswitches but not in other types or classes of the preQ₁ riboswitch family. Our data suggest that the level of regulation attained is similar between the preQ₁-I₁ Can riboswitch and the preQ₁-II Lrh riboswitch, despite differences in preQ₁ binding stoichiometry^{37, 48} (Fig. 3d). This result suggests that these two disparate riboswitch folds evolved equally effective chemical networks to sense a common effector for gene regulation. Yet, cooperativity is expected to provide notable benefits in regulation efficiency. One such advantage is that gene expression is

permitted when metabolite levels are low (Fig. 3b, middle panel), while assuring the ability to quickly attenuate expression before excess effector accrues in the cell⁴⁶. This is reasonable considering that many preQ₁-I₁ riboswitches control the translation of transporters that salvage Q-precursor metabolites from the extracellular environment^{23, 28, 49}.”

3. In the case of this example, the SDS is partially embedded within the aptamer domain. However, in Class III preQ₁ switches, the SDS is found further outside. Do the authors think such dual binding events as observed here are uniquely effective when the SDS is within the aptamer? This might be worth adding a sentence or two in the discussion.

The reviewer raises a very interesting point. We have added the following brief paragraph on Lines 252-264 that considers this possibility. We also took the last sentence from the previous version of the manuscript and added it here to close the paragraph.

“Although the Can and Lrh riboswitches differ in terms of binding stoichiometry and overall fold, each positions its expression platform near the binding pocket. This organization raises the question of whether dual, stacked metabolite binding could be effective to regulate folds in which the aptamer is located distally from the expression platform. PreQ₁-III riboswitches exemplify this organization, wherein the expression platform can be as far as 40 Å away from the aptamer³⁸. Communication between the single-effector pocket and an orthogonal SDS-antiSDS helix is mediated by an A minor base that makes a T-shaped contact with the edge of preQ₁³⁸ (Supplementary Fig. 4e). The preQ₁-II riboswitch uses a similar pocket³⁷ and the A-minor base was shown to be essential for gene-regulatory function⁴⁰. It is conceivable that dual, stacked effector recognition could be used by the preQ₁-III riboswitch fold, if the effectors promoted coaxial helical stacking, and one or both were detected by an A-minor motif. Accordingly, we predict that additional riboswitches that bind dual, stacked effectors exist in nature.”

We also modified the figure legend of **Supplementary Figure 4e** to support the A-minor statements added in the main text.

“A70 and A84 are inclined A-minor bases that originate from an orthogonal A-form helix that abuts the effector edge^{8, 9}. In the preQ₁-II riboswitch, these bases are important for gene regulation and dynamics¹⁰⁻¹².”

To be clear, these very minor comments do not dampen enthusiasm for what I consider to be a very strong and rigorous study. I strongly support the publication of this manuscript once the issues above have been addressed.

We thank the reviewer for this clarification and appreciate the thoughtful review. We thank the other reviewers as well.

**A small RNA that cooperatively senses two stacked metabolites in one pocket for
gene control**

Griffin M. Schroeder^{1,2}, Chapin E. Cavender^{1,2}, Maya E. Blau³, Jermaine L. Jenkins^{1,2}, David H. Mathews^{1,2} and Joseph E. Wedekind^{1,2}

¹ *Department of Biochemistry & Biophysics, University of Rochester School of Medicine & Dentistry, Rochester, NY 14642, USA.*

² *Center for RNA Biology, University of Rochester School of Medicine & Dentistry, Rochester, NY 14642, USA.*

³ *University of Rochester, 120 Trustee Road, Rochester, NY 14627, USA.*

*To whom correspondence should be addressed. Tel: +1 585 273 4516;
Email: joseph.wedekind@rochester.edu*

ORCID's: 0000-0001-6354-752X (GMS), 0000-0002-5899-7953 (CEC), 0000-0002-8948-4982(MEB), 0000-0003-2548-3275 (JLJ), 0000-0002-2907-6557 (DHM) and 0000-0002-4269-4229 (JEW)

27 **Abstract**

[revised manuscript text omitted]

*To whom correspondence should be addressed. Tel: +1 585 273-4516;
Email: joseph.wedekind@rochester.edu*

Supplementary Table 1: Data collection and refinement statistics (molecular replacement)

	PreQ ₁ Bound Can (PDB Entry 7rex)
Data collection	
Space group	P 3 ₂ 2 1
Cell dimensions	
a = b , c (Å)	57.8, 153.6
α = β , γ (°)	90, 120
Resolution (Å)	35.8 – 2.60 (2.72 – 2.60)*
R _{merge} (%)	9.6 (88.0)
R _{P.I.M.} (%)	7.1 (66.7)
I / σ (I)	8.6 (2.0)
Completeness (%)	99.6 (99.1)
Redundancy	4.7 (4.8)
CC1/2	0.99 (0.86)
Refinement	
Resolution (Å)	35.8 – 2.60 (2.72 – 2.60)*
No. reflections	17418
R _{work} / R _{free}	0.232/0.272
No. atoms	
RNA	2042
preQ ₁ /ion	78/4
water	1
B -factors (Å ²)	
RNA	87
preQ ₁	52
r.m.s. deviations from ideal	
Bond lengths (Å)	0.005
Bond angles (°)	0.849
clash score per 1000 atoms	2.5

*Parenthetical values indicate data in the highest resolution shell.

Supplementary Table 2: Average Thermodynamic Parameters for the Wildtype Type I PreQ₁-I Riboswitches

Sequence	$K_{D, app}$ (nM) ^a	N	ΔH_1 (kcal mol ⁻¹)	$-T\Delta S$ (kcal mol ⁻¹)	ΔG (kcal mol ⁻¹)
C. antarcticus	32.0 ± 2.0	1.8 ± 0.01	-25.5 ± 0.2	15.3 ± 0.3	-10.3 ± 0.1
H. influenzae ^b	52.9 ± 0.2	2.2 ± 0.03	-25.6 ± 1.1	15.6 ± 1.0	-10.0 ± 0.1
N. gonorrhoeae ^b	50.5 ± 1.3	2.2 ± 0.06	-21.8 ± 1.0	11.8 ± 1.0	-10.0 ± 0.1

^a Measured at 25 °C.

^b Classified as type I preQ₁-I riboswitch based on Roth *et al*¹

Supplementary Table 3: Average Thermodynamic Parameters for WT and Mutant Type I PreQ₁ Riboswitches

Parameter	Can^a WT	Can^b C17U	Can^b C31U	Hin^c WT	Ngo^d WT
ΔH_{A1} (kcal mol ⁻¹)	-36.9 (-37.8, -36.3) ^e	-1.6 (-1.8, 0.2)	8.6 (7.6, 9.4)	-40.0 (-40.4, -39.6)	-35.4 (-36.3, -34.6)
ΔH_{B1} (kcal mol ⁻¹)	-29.9 (-31.4, -28.6)	0.4 (-1.9, 0.8)	-17.7 (-18.7, -15.4)	-26.0 (-26.3, -25.8)	-31.0 (-31.0, -30.9)
ΔH_{A2} (kcal mol ⁻¹)	-47.4 (-48.8, -45.7)	-4.3 (-4.9, -2.3)	-23.5 (-25.6, -22.4)	-55.1 (-55.5, -54.7)	-37.6 (-39.0, -36.2)
ΔH_{B2} (kcal mol ⁻¹)	-40.3 (-40.9, -39.5)	-2.3 (-4.4, -2.2)	-49.7 (-51.3, -48.0)	-41.2 (-41.5, -40.9)	-33.2 (-34.7, -31.7)
$K_{D,A1}$ (nM)	1480.0 (1190.0, 1840.0)	5690.0 (5200.0, 6190.0)	9660.0 (8640.0, 11050.0)	3710.0 (3380.0, 4020.0)	1402.0 (1190.0, 1670.0)
$K_{D,B1}$ (nM)	2260.0 (1300.0, 3650.0)	6950.0 (6210.0, 8180.0)	21300.0 (17700.0, 25300.0)	9320.0 (8660.0, 9960.0)	15500.0 (4920.0, 30900.0)
$K_{D,A2}$ (nM)	182.0 (126.0, 273.0)	587.0 (492.0, 803.0)	3200.0 (2430.0, 4180.0)	113.0 (107.0, 120.0)	13.0 (7.0, 39.0)
$K_{D,B2}$ (nM)	278.0 (249.0, 305.0)	716.0 (605.0, 981.0)	7020.0 (6560.0, 7400.0)	284.0 (266.0, 307.0)	142.0 (120.0, 165.0)
C^f	8.1 (4.5, 14.3)	9.7 (7.3, 11.2)	3.0 (2.6, 3.6)	32.8 (29.7, 35.8)	109.0 (31.0, 249.0)
K_{D1}^g (nM)	891.0 (630.0, 1208.0)	3130.0 (3030.0, 3220.0)	6640.0 (6420.0, 6830.0)	2650 (2470.0, 2820.0)	1280.0 (974.0, 1580.0)
K_{D2} (nM)	461.0 (380.0, 565.0)	1300.0 (1140.0, 1750.0)	10260.0 (9460.0, 10980.0)	398.0 (379.0, 420.0)	156.0 (127.0, 200.0)
γ^h	7.7 (4.5, 12.6)	9.6 (7.2, 11.0)	2.6 (2.5, 2.7)	26.7 (23.6, 29.6)	32.9 (19.6, 49.1)

^a *Carnobacterium antarcticus* (Can)^b ITC was recorded at 25 °C; all other measurements were recorded at 37 °C.^c *Haemophilus influenzae* (Hin)^d *Neisseria gonorrhoeae* (Ngo)^e Values reported are the median and confidence intervals (2.5, 97.5) from bootstrapping (See **Methods**)^f The ratio of microscopic binding constants yields the cooperativity constant C, which shows positive cooperativity when greater than unity.^g For simplicity, the microscopic binding constants can be used to generate the macroscopic binding constants, K_{D1} and K_{D2} , corresponding to the first and second ligand binding steps (**Supplemental Fig. 5b**).^h The ratio of macroscopic binding constants (K_{D1} , K_{D2}) multiplied by a statistical factor of 4 yields the macroscopic cooperativity constant γ as described in the Methods.

Supplementary Table 4: EC₅₀ and fold change in preQ₁-induced reporter-gene repression

Riboswitch Sequence	EC _{50,1} (nM)	EC _{50,2} (nM)	EC ₅₀ Fold Change	Fold Repression
C. antarticus WT ^a	96 ± 14	7100 ± 360	N/A	15.4 ± 1.5
L. rhamnosus WT ^b	15 ± 0.1	N/A	N/A	14.9 ± 1.2
C. antarticus C17U ^b	4.3 × 10 ⁵ ± 1	N/A	60 ± 3	5.9 ± 0.7
C. antarticus C31U ^b	1.5 × 10 ⁶ ± 1	N/A	210 ± 10	1.9 ± 0.1

^a Fit with a biphasic model (see **Methods**).

^b Fit with log(inhibitor) vs response (three parameters) (see **Methods**).

Supplementary Figure 1 | Covariation model and multisequence alignments of preQ₁ class I riboswitches. (a) Type I Covariation models generated from the full group of known sequence representatives (adapted from ref. 2); red, black and gray positions indicate 97%, 90% and 75% sequence conservation. Multisequence alignments were generated using a handful of representatives derived from phylogenetically diverse bacteria (reported by McCown *et al.*²). Positions in bold within the alignment recognize preQ₁ based on the *C. antarcticus* co-crystal structure of this investigation. PreQ₁ binding nucleobases at the α and β sites are each denoted in the covariation model and the sequence alignment as α or β . Here and elsewhere, bolded organisms have been structurally characterized (this work). In addition to the greatest number of representative sequences (indicated in italics), preQ₁-I riboswitches exhibit the greatest taxonomic diversity². (b) same as (a), but with type II sequences. Characterized sequences are *T. tencongensis*^{3, 4} and *B. subtilis*⁵. Asterisks denote conserved preQ₁ recognition positions. (c) same as (a) and (b) but with type III sequences. Due to a lack of structural characterization, the canonical specificity base is the only predicted preQ₁ recognition position². Alignments were created in JALVIEW⁶.

Supplementary Figure 2 | Structural quality and details of the *C. antarcticus* preQ₁-I type I riboswitch aptamer and expression platform. (a) Reduced bias $2mF_o - DF_c$ electron-density maps contoured at 1.2σ around each riboswitch chain in the asymmetric unit. Chains A and B show electron density bathing the entire model, while chain C reveals a break at the junction between P1 and the L3 loop. Individual nucleotides are shown as a cartoon diagram for simplicity. Here and elsewhere, preQ₁ is depicted as a ball-and-stick model (green shades in chain A). **(b)** Feature-enhanced, composite-omit map showing the quality of preQ₁ models fit to unbiased electron density⁷. **(c)** The expression platforms of chains A and C form a crystal contact that likely takes the place of an intramolecular WC pair between C10 and G34.

Supplementary Figure 3 | A-amino kissing interactions between Loop 3 and the minor groove of P1. (a) View of interactions near the base of P1 where the stem transitions to loop L3. Adenines 24, 25 and 26 engage in sugar edge interactions to P1 via their WC faces and Hoogsteen edges. (b) View of the floor of the α -preQ₁ binding site. Adenines 27, 28 and 29 use their WC faces to pair with the sugar edges of P1 nucleotides, supporting the floor of the ligand binding pocket.

Supplementary Figure 4 | Binding pockets of the *Can* preQ₁-I₁ riboswitch of this investigation compared to other preQ₁ riboswitches. (a) The α preQ₁ binding site of the *Can* riboswitch. Like known class I structures in (b) and (c), preQ₁ sensing at the α site occurs by canonical *cis* Watson-Crick (WC) pairing. (b) Binding pocket of the *B. subtilis* (*Bsu*) class I type II preQ₁ riboswitch⁵. Effector readout at this site is equivalent to the *Can* riboswitch α site. (c) The binding pocket of the *T. tengcongensis* (*Tte*) class I type II preQ₁ riboswitch⁴ is equivalent to b. (d) The β site of the *Can* riboswitch of this investigation displays a new mode of preQ₁ recognition of the minor-groove edge equivalent by hydrogen bonds from U7 and C31, whereas the WC face of the ligand is open. (e) PreQ₁ binding pockets of the class II riboswitch from *L. rhamnosus* (*Lrh*)⁸ (left) and the class III riboswitch from *F. prausnitzii* (*Fpr*)⁹ (right). Despite adopting different global folds, both riboswitches use a similar constellation of ten bases for ligand recognition that involves *trans* WC interactions with specificity base C30 or C7. A70 and A84 are inclined A-minor bases that originate from an orthogonal A-form helix that abuts the effector edge^{8, 9}. In the preQ₁-II riboswitch, these bases are important for gene regulation and dynamics¹⁰⁻¹².

Supplementary Figure 5 | Representative ITC thermograms and corresponding fits: (a) Replicate thermograms and global fit of WT *C. antarcticus* (*Can*) riboswitch at 37 °C. Here and elsewhere, the gray areas represent 95% confidence intervals. (b) Schematic diagram of the two-interdependent-sites binding model used to fit cooperative isotherms of this investigation (See **Methods**). (c-d) Replicate thermograms and global fits of *Can* riboswitch mutants C17U (c) and C31U (d) at 25 °C. (e-f) Representative thermograms of WT *H. influenzae* (*Hin*) (e) and *N. gonorrhoeae* (*Ngo*) (f) performed at 25 °C and fit with an independent sites (single set of sites) binding model (Malvern Panalytical, Inc). Thermodynamic parameters are listed in **Supplementary Table 2**. (g-h) Replicate thermograms and global fits of WT *Hin* (g) and *Ngo* (h) riboswitches at 37 °C. Cooperative thermograms (a, c, d, g & h) were analyzed using a non-linear least-squared minimization fitting model developed in our lab (see **Methods**). Thermodynamic parameters for a, c, d, g & h are in **Supplementary Table 3**. Additional experimental parameters for e & f are listed in **Supplementary Table 2**.

Supplementary Figure 6 | Secondary Structure Models of Known Class I Riboswitches. (a) Type I riboswitch sequences from *C. antarcticus* (*Can*), *N. gonorrhoeae* (*Ngo*) and *H. influenzae* (*Hin*). The latter two sequences were identified previously¹. The secondary structure diagram of the *Can* riboswitch was derived from the co-crystal structure of this investigation. Here and elsewhere, positions that contact preQ₁ are boxed. Color-codes correspond to specific pseudoknot base pairing (P) and loop (L) sequences as defined¹³. (b) Structurally characterized type II aptamers that bind one preQ₁ equivalent are from *T. tengcongensis*^{3, 4, 14, 15} (*Tte*) and *B. subtilis* (*Bsu*)^{1, 5}.

Supplementary Figure 7 | Solvent accessibility of bound effectors. Bound *C. antarcticus* preQ₁-I₁ riboswitch with the atomic surface shown (semi-transparent gray). The underlying RNA is colored as in **Fig. 1a** with backbone shown as a cartoon for simplicity. Atoms in each preQ₁ effector are colored by solvent-accessible surface area (SASA).

Supplementary Figure 8 | Fluorescence emission for GFPuv constructs in live bacteria.

Normalized fluorescence emission for each riboswitch construct shows fluorescence emission in the absence of preQ₁ for all constructs except the negative control (red). Changes in fluorescence from GFPuv in the absence and presence of saturating preQ₁ (3 μM) resulted in significantly decreased emission when the reporter gene was under control of WT *L. rharnosus* (*Lrh*, $p = 0.0006$) or *C. antarcticus* (*Can*, $p = 0.0037$) riboswitches; partial reduction of GFPuv emission was observed for *Can* mutants C17U ($p = 0.017$) and C31U ($p > 0.05$). The positive and negative controls were unaffected by preQ₁. The mean and standard error of the mean (S.E.M.) are reported ($n = 3$). Significance was determined by a Student's *t*-test with Welch's correction ($*p = 0.05$, $**p = 0.005$, $***p = 0.001$).

References

1. Roth, A. et al. A riboswitch selective for the queuosine precursor preQ₁ contains an unusually small aptamer domain. *Nat. Struct. Mol. Biol.* **14**, 308-17 (2007).
2. McCown, P. J., Liang, J. J., Weinberg, Z. & Breaker, R. R. Structural, functional, and taxonomic diversity of three preQ₁ riboswitch classes. *Chem. Biol.* **21**, 880-889 (2014).
3. Jenkins, J. L., Krucinska, J., McCarty, R. M., Bandarian, V. & Wedekind, J. E., Comparison of a preQ₁ riboswitch aptamer in metabolite-bound and free states with implications for gene regulation. *J. Biol. Chem* **286**, 24626-37 (2011).
4. Schroeder, G. M. et al. Analysis of a preQ₁-I riboswitch in effector-free and bound states reveals a metabolite-programmed nucleobase-stacking spine that controls gene regulation. *Nucleic Acids Res.* **48**, 8146-8164 (2020).
5. Klein, D. J., Edwards, T. E. & Ferre-D'Amare, A. R., Cocrystal structure of a class I preQ₁ riboswitch reveals a pseudoknot recognizing an essential hypermodified nucleobase. *Nat. Struct. Mol. Biol.* **16**, 343-344 (2009).
6. Waterhouse, A. M., Procter, J. B., Martin, D. M. A., Clamp, M. & Barton, G. J., Jalview Version 2—a multiple sequence alignment editor and analysis workbench. *Bioinformatics*, 1189-1191 (2009).
7. Afonine, P. V. et al. FEM: feature-enhanced map. *Acta Crystallogr. D* **71**, 646-666 (2015).
8. Liberman, J. A., Salim, M., Krucinska, J. & Wedekind, J. E. Structure of a class II preQ₁ riboswitch reveals ligand recognition by a new fold. *Nat. Chem. Biol.* **9**, 353-355 (2013).
9. Liberman, J. A. et al. Structural analysis of a class III preQ₁ riboswitch reveals an aptamer distant from a ribosome-binding site regulated by fast dynamics. *Proc. Natl. Acad. Sci. U. S. A.* **112**, E3485-94 (2015).
10. Dutta, D. & Wedekind, J. E., Nucleobase mutants of a bacterial preQ₁-II riboswitch that uncouple metabolite sensing from gene regulation. *J. Biol. Chem.* **295**, 2555-2567 (2020).
11. Soulière, M. F. et al. Tuning a riboswitch response through structural extension of a pseudoknot. *Proc. Natl. Acad. Sci. U. S. A.* **110**, E3256–E3264 (2013).
12. Kang, M., Eichhorn, C. D. & Feigon, J. Structural determinants for ligand capture by a class II preQ₁ riboswitch. *Proc. Natl. Acad. Sci. U. S. A.* **111**, E663-E671 (2014).
13. Peselis, A. & Serganov, A. Structure and function of pseudoknots involved in gene expression control. *Wiley Interdiscip. Rev. RNA* **5**, 803-822 (2014).
14. Spitale, R. C., Torelli, A. T., Krucinska, J., Bandarian, V. & Wedekind, J. E. The structural basis for recognition of the preQ₀ metabolite by an unusually small riboswitch aptamer domain. *J Biol. Chem.* **284**, 11012-11016 (2009).

15. Kang, M.; Peterson, R. & Feigon, J. Structural Insights into riboswitch control of the biosynthesis of queuosine, a modified nucleotide found in the anticodon of tRNA. *Mol. Cell* **33**, 784-90 (2009).

REVIEWERS' COMMENTS

Reviewer #1 (Remarks to the Author):

In this revised manuscript, the authors have provided a detailed set of point-by-point responses with associated changes to the manuscript, where appropriate. In light of the comments raised by the reviews, the authors have reconsidered their analysis and interpretation of some of their data, such as the cell-based reporter assays, and provided additional information such as the phylogenetic alignment to provide further information about this novel subclass of preQ1 riboswitches. None of these changes significantly altered the original findings or conclusions, but rather serve to clarify key observations and conclusions. These revisions, in the opinion of this reviewer, have fully addressed all of the concerns raised and the current form of the manuscript is suitable for publication.

Reviewer #2 (Remarks to the Author):

All my concerns were addressed appropriately. The very colorful, as in full of colors, response addresses well all previous points raised.

Reviewer #3 (Remarks to the Author):

Thank you to the authors for responding to our comments. The authors have addressed our concerns and I think revised manuscript is suitable for publication.